# Influence of the Type of Amino Acid on the Permeability and Properties of Ibuprofenates of Isopropyl Amino Acid Esters

**DOI:** 10.3390/ijms23084158

**Published:** 2022-04-09

**Authors:** Paula Ossowicz-Rupniewska, Joanna Klebeko, Ewelina Świątek, Karolina Bilska, Anna Nowak, Wiktoria Duchnik, Łukasz Kucharski, Łukasz Struk, Karolina Wenelska, Adam Klimowicz, Ewa Janus

**Affiliations:** 1Department of Chemical Organic Technology and Polymeric Materials, Faculty of Chemical Technology and Engineering, West Pomeranian University of Technology in Szczecin, Piastów Ave. 42, PL-71065 Szczecin, Poland; joanna.klebeko@gmail.com (J.K.); ewelina.swiatek@zut.edu.pl (E.Ś.); karolinabilska99@gmail.com (K.B.); ejanus@zut.edu.pl (E.J.); 2Department of Cosmetic and Pharmaceutical Chemistry, Pomeranian Medical University in Szczecin, Powstańców Wielkopolskich Ave. 72, PL-70111 Szczecin, Poland; anowak@pum.edu.pl (A.N.); lukasz.kucharski@pum.edu.pl (Ł.K.); adam.klimowicz@pum.edu.pl (A.K.); 3Department of Pharmaceutical Chemistry, Pomeranian Medical University in Szczecin, Powstańców Wielkopolskich Ave. 72, PL-70111 Szczecin, Poland; wiktoria.duchnik@pum.edu.pl; 4Department of Organic and Physical Chemistry, Faculty of Chemical Technology and Engineering, West Pomeranian University of Technology, Al. Piastów 42, PL-71065 Szczecin, Poland; lukasz.struk@zut.edu.pl; 5Department of Nanomaterials Physicochemistry, Faculty of Chemical Technology and Engineering, West Pomeranian University of Technology in Szczecin, Piastów Ave. 45, PL-70311 Szczecin, Poland; karolina.wenelska@zut.edu.pl

**Keywords:** amino acid ionic liquids, ibuprofen, nonsteroidal anti-inflammatory medication, transdermal drug delivery, skin barrier

## Abstract

Modifications of (*RS*)-2-[4-(2-methylpropyl)phenyl] propanoic acid with amino acid isopropyl esters were synthesised using different methods via a common intermediate. The main reaction was the esterification of the carboxyl group of amino acids with isopropanol and chlorination of the amino group of the amino acid, followed by an exchange or neutralisation reaction and protonation. All of the proposed methods were very efficient, and the compounds obtained have great potential to be more effective drugs with increased skin permeability compared with ibuprofen. In addition, it was shown how the introduction of a modification in the form of an ion pair affects the properties of the obtained compound.

## 1. Introduction

Currently, the most commonly used drugs for combating pain of various origin, fever, and inflammation are included in the nonsteroidal anti-inflammatory drugs (NSAIDs) group. It is estimated that 30–50 million people daily use pharmacological agents containing medicinal substances belonging to NSAIDs. This is attributed to their broad spectrum of activity, but above all, their availability. Most of the known preparations are available on the market without a prescription. NSAIDs are a large and diverse group of compounds in terms of chemical structure, showing an acidic nature. They include compounds that are medium-strength acids (pKa 3–5). Most of this group of drugs are derivatives of carboxylic and enol acids with free functional groups of an acidic nature. Analgesic, antipyretic, and anti-inflammatory effects characterise NSAIDs. In connection with their mechanism of action and acidic character, they can cause a number of side effects, especially concerning the gastrointestinal tract, such as gastrointestinal bleeding, reduction in the volume of gastric juices, and the associated risk of ulceration, nausea, vomiting, diarrhoea, or side effects concerning the circulatory system, such blood clots and severe bronchospasm and attacks of asthmatic cough [1,2,3,4,5].

In order to avoid side effects, e.g., ulcerogenic effects related to the inhibition of PGI_2_ biosynthesis, many preparations have been introduced in which the active group’s acidic ones are obtained after enzymatic hydrolysis. After oral administration, drugs of this type pass through the stomach practically unchanged. These are prodrugs, esters (e.g., bonorylate), or other structures that allow the formation of an acidic moiety in the process of biotransformation (e.g., oxidation). There are new prodrugs with improved pharmacological activity and a lower risk of gastrointestinal side effects, such as ibuprofen amides formed by the reaction of the ibuprofen carboxyl group with heteroaromatic amines [6], ibuprofen ester derivatives obtained by the reaction of the ibuprofen carboxyl group with menthol, thymol and eugenol [7], or ibuprofen esters obtained as a result of the reaction of the ibuprofen carboxyl group with compounds such as quercetin, salicylic alcohol, or gallic acid [8].

Ibuprofen ((*RS*)-2-(4-(2-methylpropyl)phenyl)propanoic acid) is one of the most widely used NSAIDs that is commonly used in the treatment of rheumatoid arthritis and osteoarthritis, as well as for relieving pain and fever [9]. Today, it is one of the most widely used over-the-counter (OTC) medications. The most common route of administration of this drug is the oral route (it is available as tablets, capsules, and paediatric suspensions). In the treatment of pain and fever, the drug should be effective as quickly as possible, which is directly related to achieving its appropriate concentration in the blood (therapeutic dose) [10]. The rate of ibuprofen uptake into the blood depends on the dissolution rate in the body fluids. Ibuprofen is a relatively high lipophilic carboxylic acid (determined log P in the range of 2.41–4.00), with low water solubility (0.076 g dm^−3^ at 25 °C) and low permeation through the skin [11]. The solubility of ibuprofen depends on the pH and is particularly low in the stomach’s acidic environment, which requires high doses. This results in serious risk of side effects, i.e., damage to the gastric mucosa due to the acidic nature of the drug. In the pH range 1–4, the changes in ibuprofen solubility are small [12,13,14]. The increase in the solubility of ibuprofen starts from the moment of reaching the pKa. After it is reached, the solubility increases significantly, which is caused by the molecule’s ionisation under these conditions [14,15]. For example, the solubility of ibuprofen at 37 °C is 0.038 mg·cm^−3^ (pH 1), 0.084 mg·cm^−3^ (pH 4.5), 0.685 mg·cm^−3^ (pH 5.5), and 3.37 at pH 6.8 it is mg·cm^−3^ [13].

The most straightforward modification of ibuprofen is the formation of its salt. In polar solvents, including water, the solubility increases significantly by replacing an acid proton with a sodium cation [16]. Potassium, calcium, magnesium, copper, and zinc salts are also known. This has a measurable impact on faster action, including more rapid pain relief [17,18,19,20].

Increasingly, the metal cation in acid-derived drugs is replaced with an organic cation due to limitations related to solubility, bioavailability, and the possibility of introducing metallic salts into finished drug forms. Lysine ibuprofenate, obtained by combining (*RS*)-2-(4-(2-methylpropyl)phenyl)propanoic acid and lysine in a molar ratio of 1:1.14 is also known [21]. This compound is part of the commercial preparation of Lizymax. This new ibuprofen derivative is absorbed much faster after oral administration and achieves the maximum plasma concentration 3 times faster than ibuprofen acid. It is also known from the publication by Furukawa et al. that ibuprofen salt, in which the cation is derived from L-proline, is obtained by the reaction of L-proline ethyl ester and ibuprofen [22].

There have been reports that the reduction in gastrointestinal side effects can be achieved by converting the active substance into a prodrug using amino acids that are non-toxic and that additionally have a therapeutic effect on gastric damage caused by NSAIDs. For example, amides formed in the reaction of proline and ibuprofen are known, which show anti-inflammatory and neuroprotective effects while being well-tolerated by the gastric wall. In addition, L-cysteine ethyl ester ibuprofenate is known, which has a potent anti-inflammatory and antioxidant effect and reduced side effects on the gastrointestinal tract [23,24,25,26,27,28].

The topical application of drugs is one of the most important methods of delivering them to the body. A significant factor in limiting penetration is the skin barrier which reduces the penetration efficiency and limits the absorption of the compounds. This layer is the greatest obstacle to the transport of active substances and is considered the primary barrier to the permeation of molecules. It is mainly made of ceramides, cholesterol, fatty acids, cholesterol esters, and minor phospholipids. Among the available and topical medications, a significantly small group can passively cross the skin barrier in amounts sufficient to obtain a therapeutic effect. Ibuprofen is the active ingredient in many topical formulations. This route of administration is used to reduce undesirable side effects, avoid first-pass metabolism in the liver, and minimise gastrointestinal side effects [29]. However, it is difficult to obtain an effective concentration due to its poor penetration capacity through the *stratum corneum* (SC). It is related to the construction of the SC. SC consists mainly of lipids—namely, ceramides, cholesterol, and fatty acids [30].

In our previous work, we presented the greater permeation compared with pure IBU of the ion pairs of ibuprofen with L-valine alkyl esters [ValOR][IBU], in which the alkyl chain R was extended from C1 to C8 [11,31]. In the presented study, we decided to extend the modifications of IBU about other amino acids and then estimate the penetration of new compounds as well as the amount of their accumulation in the skin. Penetration in vitro was investigated using Franz diffusion cells using abdomen porcine skin. This work presents the synthesis, complete identification, and characterisation of new salts: amino acid isopropyl ester ibuprofenates [AAOR][IBU].

## 2. Results

Our previous work investigated the effect of the amino acid alkyl ester chain length on the properties of the obtained amino acid alkyl ester ibuprofenates. It has been shown that to obtain a compound with good skin permeability and good solubility in water and buffers solutions while having high biodegradability and medium lipophilicity, using an isopropyl chain is most advantageous [11,31]. As part of the research, 14 ibuprofenates of isopropyl esters of protein amino acids were obtained and tested. The studies used glycine, L-alanine, L-valine, L-isoleucine, L-leucine, L-serine, L-threonine, L-cysteine, L-methionine, L-aspartic acid, L-lysine, L-phenylalanine, and L-proline. Unfortunately, the remaining derivatives of protein amino acids could not be obtained using the presented methodology. Interestingly, the hydrochloride of the L-aspartic acid ester was obtained in the reaction of L-asparagine with isopropanol and a chlorinating agent.

As a result of the three-step reaction, the ibuprofenates of amino acid alkyl esters were obtained in high yields (85–99%). In the last step of the reaction, the amino acid isopropyl ester was combined with ibuprofen without using a solvent. The use of the solvent-free method is associated with the limitation of volatile organic compounds and the costs of the process itself. Furthermore, due to the last reaction step, no reaction by-products are formed, and the product is obtained with high purity.

All compounds obtained were identified by ^1^H and ^13^C NMR, FTIR, elemental analysis, and UV-Vis. The individual results are presented in the Appendix A. In addition, a comparison of the ^1^H NMR spectra for ibuprofen and all obtained ibuprofenates is shown in Figure 1. Significant from the point of view of confirming the ionic structure of the compounds obtained was to prove, among other things, protonation of the amino group of the amino acid. As is shown in Figure 2, the value of the NH_3_ signals shifts ranges from 4.67 ppm (for the L-methionine derivative) to 9.13 ppm (for the L-proline derivative). The integration of these signals clearly confirmed the obtained ionic structure. 

Another confirmation of obtaining the ionic structure of the ibuprofenates of isopropyl esters of amino acids was signal shifts of the carbonyl carbon of ibuprofen from 0.83 ppm (for [L-ThrOiPr][IBU]) to 5.04 ppm (for [L-CysOiPr][IBU]) in comparison with the value for this carbon in unmodified ibuprofen (181.16 ppm) [11,32,33,34]. Figure 2 presents the ^13^C NMR spectra for ibuprofen and the obtained ibuprofenates; signals of the ibuprofen carbonyl group are marked in red square and the signals of the carbonyl group in green square from the amino acid part. ^1^H and ^13^ C NMR spectra are in the Appendix A.

The characteristic IR absorption bands also confirmed the ionic structure. The FTIR spectra for ibuprofen and obtained ibuprofenates are compared in Figure 3. Two characteristic absorption bands are visible, establishing the ionic structure of the compounds obtained—asymmetric *v* (COO^−^)_as_ and symmetric stretching vibrations *v* (COO^−^)_sym._ The difference above 200 cm^−1^ between the frequency values of these two bands confirmed the presence of the carboxylate anion COO^−^ and the ionic structure of ibuprofenates [35]. FTIR and UV-Vis spectra are in the Appendix A.

Figure 4 shows the XRD patterns for ibuprofen and ibuprofen salts. Reflections that can be attributed to the ibuprofen phase ((*RS*)-2-[4-(2-methylpropyl)phenyl]propanoic acid (PDF 96-230-0213) were identified in the ibuprofen diffraction pattern. As can be seen, the diffractograms of the derivatives obtained are different from that of the starting ibuprofen. The obtained results confirmed the crystalline nature of both unmodified ibuprofen and its derivatives. XRD patterns are in the Appendix A.

The physicochemical properties such as melting point, specific rotation, thermal stability, morphology, and solubility were determined for all the compounds obtained. Table 1 summarises the results.

The melting point was determined from the DSC curves. Table 1 summarises the results. As can be seen the salts [L-ThrOiPr][IBU] (41.3 °C), [L-Asp(OiPr)_2_][IBU] (58.4 °C), [L-CysOiPr][IBU] (68.1 °C), [L-AlaOiPr][IBU] (71.0 °C), [L-ProOiPr][IBU] (71.4 °C), and [L-MetOiPr][IBU] (74.0 °C) have a lower melting point than ibuprofen (78.6 °C). All salts except [L-LysOiPr][IBU]_2_ (105.3 °C) and [L-PheOiPr][IBU] (101.4 °C) have melting points below 100 °C. The melting point and the confirmed ionic structure allow the obtained ibuprofenates to be classified into ionic liquids [36,37,38,39,40]. As it is known, obtaining ionic liquids brings advantages such as avoiding the phenomenon of polymorphism. DSC curves are in the Appendix A.

Moreover, the specific and molar rotations were determined (Table 1). As can be seen, the specific rotation depends on the type of amino acid. In most cases, the specific rotation of the resulting salt follows the specific rotation of the starting amino acid, with the exception of [L-CysOiPr][IBU] and [L-PheOiPr][IBU] in the opposite direction. The value of specific rotation of ibuprofenates is lower than that of the starting amino acids, which results from the proportion of the amino acid part in the whole molecule. A racemic mixture of ibuprofen was used for the tests. In the case of glycine salts, the obtained salt shows the ability to rotate the plane of polarised light, which may suggest a slight excess of the S(+)-ibuprofen enantiomer.

In Figure 5, the thermal stability of ibuprofen and amino acid isopropyl ester ibuprofenates was determined and compared. All TG, DTG, and c-DTA curves are in the Appendix A. The decomposition onset temperature (T_onset_) and the temperature of the maximum rate of mass loss (T_max_) were determined. The least stable is [L-AlaOiPr][IBU] (T_onset_ = 78.3 °C), while the most stable is [L-CysOiPr][IBU] (T_onset_ = 190.2 °C), which shows the onset of decomposition at a temperature higher than ibuprofen. In other cases, the decomposition onset temperature is lower for the obtained salts than for ibuprofen (T_onset_ = 189.8 °C). As can be seen, derivatives of aliphatic amino acids show lower stability—down to temperatures below 110 °C.

The maximum weight loss rate (T_max_) temperatures were similar for ibuprofen and L-amino acid isopropyl esters ibuprofenates (Table 1). [L-ValOiPr][IBU] had the lowest T_max_, while [L-PheOiPr][IBU] was the compound with the highest T_max_ (Figure 6).

One of the most critical factors affecting solubility and bioavailability is crystal morphology (size and shape of crystals). The SEM of ibuprofen and amino acid isopropyl ester ibuprofenates crystals is given in Figure 7. The shape of unmodified ibuprofen crystals is needle-type crystals. In other cases, the formation of agglomerates is visible—all derivatives of aliphatic amino acids behave similarly and form layer agglomerates. For example, the crystals of L-methionine and L-serine derivatives are needle-shaped like ibuprofen. A layered structure can also be seen for the salt of [L-Asp(OiPr)_2_][IBU], [L-LysOiPr][IBU], and [L-LysOiPr][IBU]_2_. In contrast, [L-CysOiPr][IBU] and [L-ProOiPr][IBU] were amorphous. [L-ThrOiPr][IBU] was liquid at the measurement temperature.

As is well known, knowledge of the solubility of new drugs is essential in the drug discovery and development process. The solubility in water and organic solvents was tested to classify the compounds obtained accordingly. Due to the great importance of dissolution testing in drug discovery and development, we determined the solubility in water and typical organic solvents. The solubility data are summarised in Table 2. All L-amino acid isopropyl ester ibuprofenates were soluble in ethanol and dimethyl sulfoxide and insoluble in n-hexane. L-lysine derivatives—both mono and bis (ibuprofenate) salt—were dissolved only in ethanol and DMSO; in the other solvents used for the tests, they were insoluble.

The bioavailability and effectiveness of a drug are assessed, inter alia, by assessing the solubility of the substance in water. Therefore, the solubility parameter in water and in a buffered saline solution was determined. The results, expressed in g of compound per dm^3^ and the amount of active compound in g per dm^3^, are presented in Table 3. As can be seen, the solubility of amino acid isopropyl ester ibuprofenates was markedly higher than ibuprofen, exceeding even 66 times higher (based on the concentration of active substance). Converting the drug into a salt form is a well-used technique for increasing the water solubility of a drug. Furthermore, the highest solubility in water was obtained for [L-LysOiPr][IBU] (5.047 ± 0.060 g IBU dm^−3^), while [L-PheOiPr][IBU] had the lowest (0.297 ± 0.024 g IBU dm^−3^). A typical relationship between solubility and alkyl chain length was observed: solubility decreased with increasing chain length in the case of derivatives of aliphatic amino acids. A similar dependence was observed for the solubility in PBS (pH 7.4) buffer solution. For all the tested compounds, the solubility in PBS (pH 7.4) buffer was markedly higher than in water, but these tests were conducted at a higher temperature (32 °C for PBS (pH 7.4), 25 °C for water). In this case, the highest solubility in PBS was also obtained for [L-LysOiPr][IBU] (4.779 ± 0.061 g IBU dm^−3^), while [L-PheOiPr][IBU] also had the lowest (1.017 ± 0.001 g IBU dm^−3^).

The partition coefficient is the key parameter used to predict drug hydrophobicity and partitioning in biological systems. Table 4 summarises the results of the n-octanol-water partition coefficient determined by the shake flask method. All obtained ibuprofen and amino acid isopropyl ester ibuprofenates showed a positive log P, but they were lower compared with ibuprofen. All the salts obtained are therefore more hydrophilic. 

The lipophilicity of the obtained salts was ranked in ascending order: [GlyOiPr][IBU] < [L-AlaOiPr][IBU] < [L-SerOiPr][IBU] < [L-ThrOiPr][IBU] < [L-LysOiPr][IBU]< [L-ProOiPr][IBU] < [L-LysOiPr][IBU]_2_ < [L-ValOiPr][IBU] < [L-LeuOiPr][IBU] <[L-CysOiPr][IBU] < [L-Asp(OiPr)_2_][IBU] < [L-MetOiPr][IBU] < [L-IleOiPr][IBU] < [L-PheOiPr][IBU] < IBU.

Individual permeation profiles for ibuprofen and amino acid isopropyl ester ibuprofenates are plotted in Figure 8. The content of IBU and its derivatives in the acceptor fluid collected during 24 h permeation is summarised in Table 5. The cumulative ibuprofen mass for individual compounds, determined after 24 h of permeation, was as follows: [L-LysOiPr][IBU] > [L-SerOiPr][IBU] > [L-ThrOiPr][IBU] > [L-Asp(OiPr)_2_][IBU] > [L-ValOiPr][IBU] > [GlyOiPr][IBU] > [L-ProOiPr][IBU] > [L-PheOiPr][IBU] > [L-AlaOiPr][IBU] > [L-CysOiPr][IBU] > [L-MetOiPr][IBU] > [L-LeuOiPr][IBU] > [L-LysOiPr][IBU]_2_ > [L-IleOiPr][IBU] and [IBU]. Among the studied derivatives, [L-LysOiPr][IBU] permeated to a higher degree than others; the cumulative amount of substance permeated during the 24 h study was 528.643 ± 22.716 (Table 5, Figure 8). The formation of structural modifications of ibuprofen in the form of charged compounds with lower lipophilicity has an influence on better permeability compared with the more lipophilic ibuprofen. The cluster analysis graph shows the cumulative mass of the IBU and its derivatives measured over the entire 24 h permeation period (Appendix A). In this diagram, three distinct groups characterised by similar permeation can be distinguished (circles A, B, C). Generally, the cumulative mass of all ibuprofen derivatives in the acceptor phase after 24 h of permeation was significantly higher than free ibuprofen (Appendix A). The similarity between derivatives was found using Tukey’s test, which showed a statistically significant difference between all derivatives and pure ibuprofen (*p* < 0.05) (Appendix A).

For NSAIDs, including ibuprofen, faster permeation through the skin is preferable to achieve a rapid therapeutic effect. Fast and increased permeation causes a quicker decrease in inflammation in the underlying tissues [38]. The permeation rate determined at each time interval is presented in Figure 9. The highest permeation rate to the acceptor fluid generally was observed in samples collected between 3 and 5 h.

The results of the in vitro efficiency permeation experiments related to ibuprofen and its derivatives are summarised in Table 6. The permeation parameters, including flux (J_SS_, µg IBU cm^−2^ h^−1^), apparent permeability coefficient (K_P_∙10^3^, cm h^−1^), lag time (L_T_, h), diffusion coefficient in the skin (D∙10^4^, cm^2^ h^−1^), skin partition coefficient (K_m_), percentage drug permeated after 24 h (Q_%24 h)_, and enhancement factor (EF) were designated. A significant difference in the flux for ibuprofen and its derivatives was visible. The transdermal flux of the [AAOiPr][IBU] from PBS (pH 7.4) saturated solution was from 3.16 to 21.91 times higher than that of the saturated ibuprofen solution in PBS (pH 7.4) (9.545 for [L-IleOiPr][IBU], 66.106 for [L-SerOiPr][IBU] versus 3.017 µg IBU cm^−2^ h^−1^, respectively). In the case of using ibuprofenates, higher flows of active substance were obtained, which can be very useful in using this type of derivative in formulations administered through the skin—in the form of gels, ointments, creams, or medical patches. The permeability coefficient is a quantitative measure of the rate at which a molecule can cross the skin, composed of factors related to the drug and the barrier and their interaction. This parameter eliminates the effect of concentration. For the tested compounds, K_P_ values ranged from 1.475 (for [GlyOiPr][IBU]) to 6.839 (for [L-PheOiPr][IBU]) times higher than for ibuprofen, except for [L-IleOiPr][IBU], for which this value was 2.578 times lower. The lag time depended on the type of amino acid used for modification. The lag time was significantly reduced when using amino acids with a polar side chain, such as L-lysine, L-aspartic acid, L-threonine, and L-serine. The more hydrophobic the chain, the more lengthened the lag time. In general, some of the compounds showed a similar lag time as ibuprofen, such as [GlyOiPr][IBU], [L-ValOiPr][IBU], [L-MetOiPr][IBU], [L-PheOiPr][IBU], and [L-ProOiPr][IBU]. The diffusion coefficient in the skin was approximately 2.222 and 4.134 cm^2^ h^−1^ for [L-CysOiPr][IBU] and [L-Asp(OiPr)_2_][IBU], respectively. The equilibrium solubility of the drug in the stratum corneum concerning its solubility in the vehicle was also determined. This parameter describes the drug’s ability to escape from solution and travel to the outermost layers of the stratum corneum. Its values were generally higher than for ibuprofen (1.173). They ranged from 1.573 to 7.843 for [L-MetOiPr][IBU] and [L-PheOiPr][IBU], respectively, except for [L-IleOiPr][IBU], for which the value was lower and equalled 0.455. The enhancement factor was also determined, as seen from the data presented in Table 6. The skin permeation enhancement value is the highest for [L-PheOiPr][IBU]. The obtained results suggest that the tested compounds may promote skin permeability. The selection of the appropriate structural modification of the active compound should result from many factors, such as molar mass, solubility, or lipophilicity.

The drugs may both permeate and accumulate in the skin. In the case of anti-inflammatory drugs applied topically, faster and greater permeation is most often preferred, with less accumulation in the skin [9,29]. Figure 10 shows the mass of [IBU] and its derivatives accumulated in porcine skin after 24 h of penetration. All the compounds used accumulated in the skin. The lowest accumulation for ibuprofen derivatives values was obtained for [L-LysOiPr][IBU]_2_, [L-Asp(OiPr)_2_][IBU], and [L-SerOiPr][IBU]: 392.593 ± 111.434, 406.961 ± 17.799, and 407.652 ± 74.170 µg IBU g^−1^, respectively (Figure 10). The highest accumulation was observed for [GlyOiPr][IBU] (785.483 ± 95.862), [L-IleOiPr][IBU] (646.293 ± 123.070), and [L-ThrOiPr][IBU] (621.866 ± 112.163 µg IBU g^−1^).

## 3. Materials and Methods

The materials used for the tests and the methodologies used to assess the identity and determine the properties are presented in the Appendix A.

### 3.1. Synthesis of Amino Acids Isopropyl Ester Ibuprofenates

The general scheme of the reaction is presented in Figure 11. In the first step, the hydrochlorides of the amino acid isopropyl esters were obtained in the presence of a chlorinating agent in the reaction of amino acid with isopropanol. An excess of isopropanol was used as the reaction medium in the reaction. Trimethylsilane chloride was used as the chlorinating agent, and the reaction can be successfully carried out in the presence of thionyl chloride. Due to safety, TMSCl was used. A molar excess of the chlorinating agent was used (1:2, AA/TMSCl).

The next step was the neutralisation reaction. Again, NaOH, KOH or 25% ammonia solution can be successfully used for the reaction. The procedures for preparing hydrochlorides and esters are described in detail in the Appendix A.

The last step was the protonation of the amino group of the amino acid with acid, which results in the final product. The reaction was carried out in an organic solvent (diethyl ether or chloroform) in our previous publications [11,31]. In this publication, we present a new solvent-free reaction method. First, the appropriate amino acid isopropyl ester (1.0 mmol) was added to ibuprofen (1.0 mmol), and it was mixed for 30 min at room temperature. After the reaction had ended, the product was dried for 24 h at 50 °C under reduced pressure. All syntheses were performed three times on a 0.5–5 g of product scale. The purity and identity of obtained compounds were confirmed by NMR, FT-IR, and elemental analysis methods. The spectra of all products are presented in the Appendix A.

[GlyOiPr][IBU]—glycine isopropyl ester ibuprofenate

Glycine isopropyl ester ibuprofenate was obtained as a white solid in 98% yield.



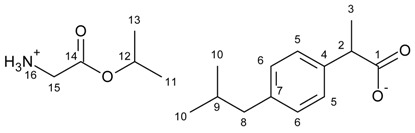



^1^H NMR (400 MHz, CDCl_3_) δ in ppm: 7.21 (d, J_6,5_ = 8.1 Hz, 2H, H5), 7.08 (d, J_5,6_ = 8.1 Hz, 2H, H6), 5.74 (s, 3H, H16), 5.05 (hept, J_11(13),12_ = 6.3 Hz, 1H, H12), 3.65 (q, J_3,2_ = 7.2 Hz, 1H, H2), 3.39 (s, 2H, H15), 2.43 (d, J_9,8_ = 7.2 Hz, 2H, H8), 1.89–1.79 (m, 1H, H9), 1.46 (d, J_2,3_ = 7.2 Hz, 3H, H3), 1.24 (d, J_12_,_11(13)_ = 6.2 Hz, 6H, H11, H13), 0.89 (d, J_9,10_ = 6.6 Hz, 6H, H10). ^13^C NMR (100 MHz, CDCl_3_) δ in ppm: 179.14 (C1), 172.48 (C14), 140.38 (C4), 138.17 (C7), 129.27 (C5), 127.22 (C6), 68.87 (C12), 45.45 (C2), 45.04 (C8), 42.85 (C15), 30.16 (C9), 22.39 (C10), 21.77 (C11, C13), 18.41 (C3). FT-IR: ν (ATR): 3162, 2983, 2957, 2930, 2867, 2605, 2471, 2237, 2227, 2220, 2209, 2199, 2190, 2177, 2166, 2164, 2160, 2152, 2071, 1745, 1624, 1560, 1487, 1466, 1393, 1384, 1362, 1288, 1246, 1178, 1144, 1106, 1056, 1022, 999, 921, 897, 880, 841, 814, 798, 740, 724, 664, 562, 543, 497, 409 cm^−1^. UV-Vis (EtOH): λ_max_ = 220.0, 265.0 nm. Elemental analysis: Calc. (%) for C_18_H_29_NO_4_ (323.427): C 66.84, H 9.04, N 4.33, O 19.79, found: C 66.83, H 9.04, N 4.33, O 19.80.

[L-AlaOiPr][IBU]—L-alanine isopropyl ester ibuprofenate

L-alanine isopropyl ester ibuprofenate was obtained as a white solid in 97% yield.



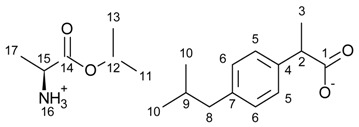



^1^H NMR (400 MHz, CDCl_3_) δ in ppm: 7.21 (d, J_6,5_ = 8.1 Hz, 2H, H5), 7.07 (d, J_5,6_ = 8.1 Hz, 2H, H6), 5.72 (s, 3H, H16), 5.08–4.94 (m, 1H, H12), 3.64 (q, J_3,2_ = 7.1 Hz, 1H, H2), 3.55 (q, J_17,15_ = 7.1 Hz, 1H, H15), 2.43 (d, J_9,8_ = 7.2 Hz, 2H, H8), 1.91–1.76 (m, 1H, H9), 1.46 (d, J_2,3_ = 7.2 Hz, 3H, H3), 1.30 (d, J_15,17_ = 7.1 Hz, 3H, H17), 1.23 (dd, J_12,11(13)_ = 6.2, 2.4 Hz, 6H, H13, H11), 0.89 (d, J_9,10_ = 6.6 Hz, 6H, H10). ^13^C NMR (100 MHz, CDCl_3_) δ in ppm: 179.15 (C1), 174.83 (C14), 140.32 (C4), 138.31 (C7), 129.25 (C5), 127.24 (C6), 68.76 (C12), 49.47 (C15), 45.57 (C2), 45.07 (C8), 30.19 (C9), 22.42 (C11), 21.73 (C10), 21.71 (C10), 19.59 (C3), 18.48 (C17). FT-IR: ν (ATR): 2954, 2921, 2954, 2867, 2848, 2708, 2644, 2546, 2191, 1749, 1706, 1617, 1568, 1528, 1452, 1422, 1377, 1362, 1338, 1327, 1310, 1289, 1227, 1176, 1148, 1120, 1105, 1062, 1038, 1022, 1006, 946, 906, 876, 862, 818, 800, 757, 728, 668, 635, 552, 508, 482, 463, 412 cm^−1^. UV-Vis (EtOH): λ_max_ = 220.0, 264.0 nm. Elemental analysis: Calc. (%) for C_19_H_31_NO_4_ (337.454): C 67.63, H 9.26, N 4.15, O 18.97, found: C 67.64, H 9.25, N 4.15, O 18.96.

[L-ValOiPr][IBU]—L-valine isopropyl ester ibuprofenate

L-valine isopropyl ester ibuprofenate was obtained as a white solid in 94% yield.



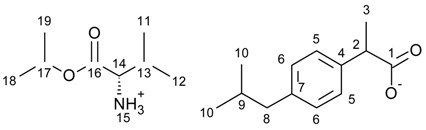



^1^H NMR (400 MHz, CDCl_3_) δ in ppm: 7.21 (d, 2H, *J*_5,6_ = 8.1 Hz, H5), 7.07 (d, 2H, *J*_6,5_ = 8.1 Hz, H6), 5.59 (s, 3H, H15), 5.01–5.07 (m, 1H, H17), 3.61–3.66 (q, 1H, H2), 3.36 (d, 1H, *J*_14,13_ = 3.9 Hz, H14), 2.43 (d, 2H, *J*_8,9_ = 7.3 Hz, H8), 2.04–2.08 (m, 1H, H13), 1.81–1.87 (m, 1H, H9), 1.45 (d, 3H, *J*_3,2_ = 7.3 Hz, H3), 1.24 (dd, 6H, *J*_19,17_ = 6.2 *J*_18,17_ = 6.2 Hz, H18, H19), 0.93 (d, 3H, *J*_12,13_ = 7.1 Hz, H12), 0.89 (d, 6H, *J*_10,9_ = 6.6 Hz, H10), 0.88 (d, 3H, *J*_11,13_ = 6.9 Hz, H11). ^13^C NMR (100 MHz, CDCl_3_) δ in ppm: 179.17 (C1), 173.34 (C16), 140.18 (C4), 138.46 (C7), 129.18 (C5), 127.25 (C6), 68.71 (C17), 58.95 (C14), 45.73 (C8), 45.06 (C2), 31.47 (C13), 30.19 (C9), 22.40 (C10), 21.78 (C18, C19), 18.73 (C12), 18.53 (C3), 17.23 (C11). FT-IR: ν (ATR): 2966, 2953, 2925, 2869, 2606, 1742, 1614, 1585, 1509, 1465, 1417, 1386, 1366, 1330, 1288, 1228, 1182, 1146, 1106, 1053, 917, 880, 825, 786, 677, 592 cm^−1^. UV-Vis (EtOH): λ_max_ = 219.3, 264.0 nm. Elemental analysis: Calc. (%) for C_21_H_35_NO_4_ (365.512): C 69.01, H 9.65, N 3.83, O 17.51, found: C 69.23, H 9.65, N 3.86, O 17.45.

[L-IleOiPr][IBU]—L-isoleucine isopropyl ester ibuprofenate

L-isoleucine isopropyl ester ibuprofenate was obtained as a white solid in 93% yield.



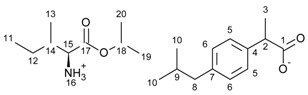



^1^H NMR (400 MHz, CDCl_3_) δ in ppm: 7.21 (d, J_6,5_ = 8.1 Hz, 2H, H5), 7.07 (d, J_5,6_ = 8.0 Hz, 2H, H6), 5.46 (s, 3H, H16), 5.04 (hept, J_19(20),18_ = 6.3 Hz, 1H, H18), 3.64 (q, J_3,2_ = 7.2 Hz, 1H, H2), 3.42 (d, J_14,15_ = 4.5 Hz, 1H, H15), 2.43 (d, J_9,8_ = 7.2 Hz, 2H, H8), 1.91–1.71 (m, 2H, H14, H9), 1.46 (d, J_2,3_ = 7.2 Hz, 3H, H3) 1.43–1.36 (m, 1H, H12′), 1.24 (dd, J_18,19(20)_ = 6.2, 3.3 Hz, 6H, H20, H19), 1.21–1.14 (m, 1H, H12”), 0.90 (dd, J = 6.8, 4.8 Hz, 12H, H11, H10, H13). ^13^C NMR (100 MHz, CDCl_3_) δ in ppm: 179.22 (C1), 173.65 (C17), 140.31 (C4), 138.24 (C7), 129.23 (C5), 127.25 (C6), 68.63 (C18), 58.18 (C15), 45.54 (C2), 45.07 (C8), 38.60 (C14), 30.20 (C9), 24.88 (C12), 22.42 (C10), 21.82 (C20, C19), 18.47 (C3), 15.36 (C11), 11.75 (C13). FT-IR: ν (ATR): 2953, 2932, 2877, 2867, 2613, 2182, 2172, 1739, 1615, 1508, 1464, 1381, 1365, 1328, 1284, 1221, 1169, 1103, 1059, 1026, 1008, 961, 941, 903, 879, 858, 825, 812, 785, 756, 730, 718, 700, 669, 639, 589, 555, 589, 525, 488, 431 cm^−1^. UV-Vis (EtOH): λ_max_ = 220.1, 265.0 nm. Elemental analysis: Calc. (%) for C_22_H_37_NO_4_ (379.533): C 69.62, H 9.83, N 3.59, O 16.86, found: C 69.63, H 9.83, N 3.58, O 16.87.

[L-LeuOiPr][IBU]—L-leucine isopropyl ester ibuprofenate

L-leucine isopropyl ester ibuprofenate was obtained as a white solid in 93% yield.



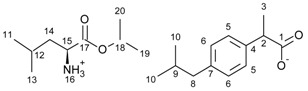



^1^H NMR (400 MHz, CDCl_3_) δ in ppm: 7.21 (d, J_6,5_ = 8.1 Hz, 2H, H5), 7.07 (d, J_5,6_ = 8.1 Hz, 2H, H6), 5.69 (s, 3H, H16), 5.02 (hept, J_11(13),12_ = 6.3 Hz, 1H, H18), 3.64 (q, J_3,2_ = 7.1 Hz, 1H, H2), 3.49 (t, J_14,15_ = 8.3, 1H, H15), 2.43 (d, J_9,8_ = 7.2 Hz, 2H, H8), 1.83 (dq, J_10,9_ = 13.6, 6.8 Hz, 1H, H9), 1.75–1.66 (m, 1H, H12), 1.54 (ddd, J = 13.8, 8.0, 5.9 Hz, 1H, H14), 1.46 (d, J_2,3_ = 7.2 Hz, 3H, H3), 1.44–1.37 (m, 1H, H14), 1.23 (dd, J = 6.3, 2.9 Hz, 6H, H19, H20), 0.91–0.87 (m, 12H, H11, H13, H19, H20). ^13^C NMR (100 MHz, CDCl_3_) δ in ppm: 179.18 (C1), 174.86 (C17), 140.30 (C4), 138.29 (C7), 129.24 (C5), 127.26 (C6), 68.68 (C18), 52.25 (C15), 45.57 (C2), 45.07 (C8), 43.23 (C14), 30.20 (C9), 24.67 (C11), 22.79 (C12), 22.42 (C19), 21.89 (C20), 21.76 (C10), 21.73 (C10), 18.48 (C3). FT-IR: ν (ATR): 2954, 2928, 2869, 2842, 1738, 1613, 1560, 1514, 1465, 1386, 1366, 1358, 1327, 1288, 1225, 1190, 1180, 1168, 1146, 1135, 1102, 1061, 908, 813, 730, 545 cm^−1^. UV-Vis (EtOH): λ_max_ = 219.7, 265.0 nm. Elemental analysis: Calc. (%) for C_22_H_37_NO_4_ (379.533): C 69.62, H 9.83, N 3.59, O 16.86, found: C 69.61, H 9.84, N 3.59, O 16.86.

[L-SerOiPr][IBU]—L-serine isopropyl ester ibuprofenate

L-serine isopropyl ester ibuprofenate was obtained as a pale yellow solid in 85% yield.



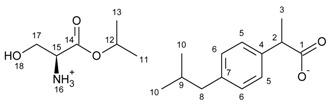



^1^H NMR (400 MHz, CDCl_3_) δ in ppm: 7.19 (d, J_6,5_ = 8.1 Hz, 2H, H5), 7.06 (d, J_5,6_ = 8.1 Hz, 2H, H6), 6.05 (s, 4H, H16, H18), 5.08–5.01 (m, 1H, H12), 3.81 (dd, J_15,17_ = 11.7, 3.8 Hz, 1H, H17), 3.72 (dd, J_15,17_ = 11.6, 5.2 Hz, 1H, H17), 3.60 (q, J_3,2_ = 7.1 Hz, 1H, H2), 3.51 (t, J_17,15_ = 5.2, 1H, H15), 2.43 (d, J_9,8_ = 7.2 Hz, 2H, H8), 1.82 (dq, J_10,9_ = 13.5, 6.8 Hz, 1H, H9), 1.43 (d, J_3,2_ = 7.1 Hz, 3H, H3), 1.25 (dd, J_12,11(13)_ = 6.3, 3.2 Hz, 6H, H11, H13), 0.89 (d, J_10,9_ = 6.6 Hz, 6H, H10). ^13^C NMR (100 MHz, CDCl_3_) δ in ppm: 180.21 (C1), 171.27 (C14), 140.19 (C4), 138.75 (C7), 129.21 (C5), 127.25 (C6), 69.65 (C12), 62.47 (C17), 55.35 (C15), 46.10 (C2), 45.03 (C8), 30.15 (C9), 22.40 (C10), 21.69 (C11, C13), 18.58 (C3). FT-IR: ν (ATR): 3173, 3050, 2979, 2951, 2921, 2866, 2759, 2652, 1746, 1629, 1560, 1558, 1518, 1463, 1411, 1397, 1374, 1364, 1321, 1293, 1248, 1237, 1162, 1145, 1111, 1094, 1071, 1058, 1039, 1008, 982, 941, 899, 883, 827, 784, 629 cm^−1^. UV-Vis (EtOH): λ_max_ = 219.7, 265.0 nm. Elemental analysis: Calc. (%) for C_19_H_31_NO_5_ (353.453): C 64.56, H 8.84, N 3.96, O 22.63, found: C 64.58, H 8.83, N 3.95, O 22.64.

[L-ThrOiPr][IBU]—L-threonine isopropyl ester ibuprofenate

L-threonine isopropyl ester ibuprofenate was obtained as a white semi-solid in 89% yield.



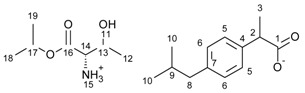



^1^H NMR (400 MHz, CDCl_3_) δ in ppm: 7.18 (d, J_5,6_ = 7.9 Hz, 2H, H5), 7.03 (d, J_6,5_ = 7.8 Hz, 2H, H6), 6.68 (s, 4H, H11, H15), 5.03 (hept, J_19,17_ = 6.3 Hz, 1H, H17), 4.00–3.89 (m, 1H, H13), 3.57 (q, J_3,2_ = 7.1 Hz, 1H, H2), 3.29 (d, J_13,14_ = 5.2 Hz, 1H, H14), 2.41 (d, J_9,8_ = 7.2 Hz, 2H, H8), 1.89–1.75 (m, 1H, H9), 1.40 (d, J_2,3_ = 7.1 Hz, 3H, H3), 1.26–1.10 (m, 6H, H19, H12), 1.17 (d, J_17,18_ = 6.4 Hz, 3H, H18), 0.88 (d, J_9,10_ = 6.6 Hz, 6H, H10). ^13^C NMR (100 MHz, CDCl_3_) δ in ppm: 180.34 (C1), 170.80 (C16), 139.87 (C4), 139.37 (C7), 129.10 (C5), 127.34 (C6), 69.73 (C17), 67.18 (C13), 59.38 (C14), 46.58 (C2), 45.07 (C8), 30.20 (C9), 22.43 (C18), 21.61 (C19), 19.96 (C10), 18.86 (C3), 18.81 (C12). FT-IR: ν (ATR): 3089, 3051, 3020, 2953, 2926, 2868, 2847, 1739, 1549, 1512, 1460, 1419, 1384, 1365, 1287, 1221, 1181, 1145, 1102, 1059, 1022, 1001, 954, 916, 896, 884, 852, 785, 754, 727, 666, 634, 593, 546, 494, 468, 446 cm^−1^. UV-Vis (EtOH): λ_max_ = 219.5, 265.0 nm. Elemental analysis: Calc. (%) for C_20_H_33_NO_5_ (367.480): C 65.37, H 9.05, N 3.81, O 21.77, found: C 65.37, H 9.04, N 3.82, O 21.76.

[L-CysOiPr][IBU]—L-cysteine isopropyl ester ibuprofenate.

L-cysteine isopropyl ester ibuprofenate was obtained as a yellowish solid in 95% yield.



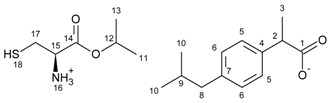



^1^H NMR (400 MHz, CDCl_3_) δ in ppm: 7.21 (d, J_5,6_ = 8.1 Hz, 2H, H6), 7.08 (d, J_6,5_ = 8.1 Hz, 2H, H5), 5.34 (s, 4H, H16, H18), 5.08–5.01 (hept, J_11(13),12_ = 6.3 Hz, 1H, H12), 3.81 (t, J_11(13),12_ = 7.3, 4.5 Hz, 2H, H12), 3.66 (t, J_16,15_ = 7.1 Hz, 1H, H15), 3.11 (dd, J_18,17_ = 13.8, 4.5 Hz, 1H, H17), 2.94 (dd, J_18,17_ = 13.8, 7.4 Hz, 1H, H17), 2.43 (d, J_9,8_ = 7.2 Hz, 2H, H8), 1.86–1.78 (m, 1H, H9), 1.46 (d, J_2,3_ = 7.2 Hz, 3H, H3), 1.25 (d, J = 6.3 Hz, 6H, H11, H13), 0.89 (d, J_10,9_ = 6.6 Hz, 6H, H10). ^13^C NMR (100 MHz, CDCl_3_) δ in ppm: δ 176.12 (C18), 173.59 (C1), 139.93 (C4), 139.14 (C7), 129.38 (C5), 127.55 (C6), 68.40 (C12), 54.24 (C15), 44.96 (C2), 44.67 (C8), 44.03 (C9), 30.07 (C17), 22.63 (C11, C12), 21.96 (C10), 19.04 (C3). FT-IR: ν (ATR): 2953, 2923, 2866, 2629, 1744, 1732, 1598, 1529, 1511, 1454, 1383, 1374, 1366, 1358, 1283, 1257, 1228, 1206, 1180, 1167, 1145, 1103, 162, 1023, 996, 911, 882, 864, 851, 813, 792, 785, 753, 693, 600 cm^−1^. UV-Vis (EtOH): λ_max_ = 225.0, 265.0 nm. Elemental analysis: Calc. (%) for C_19_H_31_NO_4_S (365.519): C 61.76, H 8.46, N 3.79, O 17.32, S 8.68, found: C 61.76, H 8.45, N 3.80, O 17.31, S 8.68.

[L-MetOiPr][IBU]—L-methionine isopropyl ester ibuprofenate.

L-methionine isopropyl ester ibuprofenate was obtained as a white solid in 98% yield.



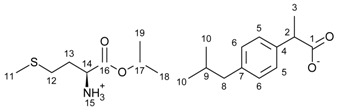



^1^H NMR (400 MHz, CDCl_3_) δ in ppm: 7.22 (d, J_5,6_ = 8.1 Hz, 2H, H5), 7.09 (d, J_6,5_ = 8.1 Hz, 2H, H6), 5.04 (hept, *J* _18(19),17_= 6.3 Hz, 1H, H17), 4.67 (s, 3H, H15), 3.67 (q, J_3,2_ = 7.1 Hz, 1H, H2), 3.59 (t, J = 7.8, 5.0 Hz, 1H, H14), 2.58 (t, J_12,13_ = 7.5, 2H, H12), 2.44 (d, J_8,9_ = 7.1 Hz, 2H, H8), 2.08 (s, 3H, H11), 2.06–1.96 (m, 1H, H13′), 1.91– 1.74 (m, 2H, H13”, H9), 1.48 (d, J_3,2_ = 7.1 Hz, 3H, H3), 1.25 (dd, J_17,18(19)_ = 6.3, 2.0 Hz, 6H, H18, H19), 0.89 (d, J_10,9_ = 6.6 Hz, 6H, H10). ^13^C NMR (100 MHz, CDCl_3_) δ in ppm: 179.23 (C1), 174.48 (C16), 140.54 (C4), 137.70 (C7), 129.30 (C5), 127.22 (C6), 68.82 (C17), 53.10 (C14), 45.18 (C2), 45.03 (C8), 33.39 (C9), 30.25 (C12), 30.16 (C13), 22.39 (C10), 21.77 (C18), 21.74 (C19), 18.29 (C3), 15.32 (C11). FT-IR: ν (ATR): 3049, 3024, 21961, 2912, 2865, 2843, 1740, 1674, 1606, 1557, 1541, 1513, 1465, 1425, 1386, 1366, 1357, 1309, 1283, 1263, 1225, 1190, 1180, 1166, 1146, 1101, 1071, 1061, 1035, 1022, 1006, 997, 967, 954, 912, 903, 877, 850, 817, 798, 729, 699, 669, 647, 636, 627, 601, 561, 544, 519, 501, 440, 430 cm^−1^. UV-Vis (EtOH): λ_max_ = 209.4, 218.0, 266.0 nm. Elemental analysis: Calc. (%) for C_21_H_35_NO_4_S (397.572): C 63.44, H 8.87, N 3.52, O 16.10, S 8.07, found: C 63.45, H 8.87, N 3.51, O 16.10, S 8.06.

[L-Asp(OiPr)_2_][IBU]—L-aspartic acid isopropyl ester ibuprofenate

L-aspartic acid isopropyl ester ibuprofenate was obtained as a white semi-solid in 99% yield.



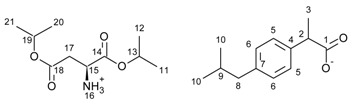



^1^H NMR (400 MHz, CDCl_3_) δ in ppm: 7.22 (d, J_5,6_ = 8.1 Hz, 2H, H5), 7.09 (d, J_6,5_ = 8.1 Hz, 2H, H6), 5.64 (s, 3H, H16), 5.11–4.94 (m, 2H, H13, H19), 3.80–3.75 (m, 1H, H15), 3.66 (q, J_3,2_ = 7.2 Hz, H2), 2.75 (dd, J_15,17_ = 5.5, 1.6 Hz, 2H, H17), 2.44 (d, J_7,8_ = 7.2 Hz, 2H, H8), 1.91–1.77 (m, 1H, H9), 1.47 (d, J_2,3_ = 7.1 Hz, 3H, H3), 1.26–1.21 (m, 12H, H11, H12, H21, H22), 0.89 (d, J_9,10_ = 6.6 Hz, H10). ^13^C NMR (100 MHz, CDCl_3_) δ in ppm: 179.12 (C1), 173.01 (C18), 170.70 (C14), 140.49 (C4), 137.91 (C7), 129.32 (C5), 127.25 (C6), 69.11 (C19), 68.49 (C13), 50.74 (C15), 45.26 (C2), 45.07 (C8), 38.03 (C17), 30.19 (C9), 22.42 (C21), 21.80 (C20), 21.78 (C12), 21.72 (C11), 21.68 (C10), 18.37 (C3). FT-IR: ν (ATR): 2978, 2954, 2932, 2869, 2147, 1728, 1587, 1512, 1466, 1373, 1327, 1233, 1205, 1179, 1146, 1104, 1064, 1021, 1000, 954, 940, 923, 896, 881, 859, 827, 802, 785, 753, 684, 668, 637, 593, 506, 479, 436, 431, 423, 419, 414, 406 cm^−1^. UV-Vis (EtOH): λ_max_ = 219.4, 264.0 nm. Elemental analysis: Calc. (%) for C_23_H_37_NO_6_ (423.543): C 65.22, H 8.81, N 3.31, O 22.67, found: C 65.23, H 8.80, N 3.32, O 22.68.

[L-LysOiPr][IBU]—L-lysine isopropyl ester ibuprofenate

L-lysine isopropyl ester ibuprofenate was obtained as a pale yellow solid in 91% yield.



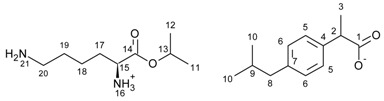



^1^H NMR (400 MHz, DMSO-*d*_6_) δ in ppm: 7.17 (d, *J*_6,5_ = 8.1 Hz, 2H, H5), 7.02 (d, *J*_5,6_ = 8.0 Hz, 2H, H6), 5.66 (s, 3H, H16), 4.90 (hept, *J*_11(12),13_ = 6.3 Hz, 1H, H13), 3.41 (q, *J_3,2_* = 7.1 Hz, 1H, H2), 3.21 (t, *J_17,15_* = 7.4, 1H, H15), 2.64 (t, *J_18,17_* = 7.5 Hz, 2H, H17), 2.39 (d, *J_9,8_* = 7.1 Hz, 2H, H8), 1.79 (dt, *J_10,9_* = 13.5, 6.7 Hz, 1H, H9), 1.57–1.30 (m, 6H, H18, H19, H20), 1.27 (d, *J*_2,3_= 7.1 Hz, 3H, H3), 1.19 (dd, *J_13,11(12)_* = 6.3, 5.0 Hz, 6H, H11, H12), 0.85 (d, *J_9,10_* = 6.6 Hz, 6H, H10). ^13^C NMR (100 MHz, DMSO-*d*_6_) δ in ppm: 177.38 (C1), 175.65 (C14), 141.57 (C4), 138.86 (C7), 128.92 (C5), 127.58 (C6), 67.68 (C13), 54.27 (C15), 47.20 (C2), 44.77 (C8), 39.16 (C20), 34.51 (C17), 30.15 (C9), 28.15 (C19), 22.73 (C18), 22.67 (C10), 22.05 (C12), 22.04 (C11), 19.97 (C3). FT-IR: ν (ATR): 3013, 2952, 2926, 2868, 2844, 2646, 1742, 1626, 1512, 1465, 1452, 1420, 1382, 1364, 1310, 1281, 1257, 1225, 1167, 1146, 1103, 1063, 1023, 997, 922, 882, 845, 819, 791, 755, 728, 718, 692, 630, 595, 525, 432 cm^−1^. UV-Vis (EtOH): λ_max_ = 227.3, 266.0 nm. Elemental analysis: Calc. (%) for C_22_H_38_N_2_O_4_ (394.548): C 66.97, H 9.71, N 7.10, O 16.22, found: C 66.98, H 9.72, N 7.11, O 16.23.

[L-LysOiPr][IBU]_2_—L-lysine isopropyl ester bis(ibuprofenate)

L-lysine isopropyl ester bis(ibuprofenate) was obtained as a pale yellow solid in 92% yield.



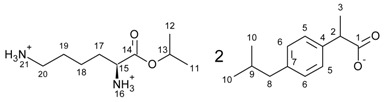



^1^H NMR (400 MHz, DMSO-*d*_6_) δ in ppm: 7.17 (d, *J*_6,5_ = 8.1 Hz, 4H, H5), 7.03 (d, *J*_5,6_ = 8.0 Hz, 4H, H6), 6.45 (s, 6H, H16, H21), 4.91 (hept, *J*_11(12),13_ = 6.3 Hz, 1H, H13), 3.46 (q, *J_3,2_* = 7.1 Hz, 2H, H2), 3.22 (t, *J_17,15_* = 7.5, 1H, H15), 2.63 (t, *J_18,17_* = 7.5 Hz, 2H, H17), 2.39 (d, *J_9,8_* = 7.1 Hz, 4H, H8), 1.80 (dt, *J_10,9_* = 13.5, 6.7 Hz, 2H, H9), 1.58–1.30 (m, 6H, H18, H19, H20), 1.28 (d, *J* _2,3_= 7.1 Hz, 6H, H3), 1.19 (dd, *J_13,11(12)_* = 6.2, 5.0 Hz, 6H, H11, H12), 0.86 (d, *J_9,10_* = 6.6 Hz, 12H, H10). ^13^C NMR (100 MHz, DMSO-*d*_6_) δ in ppm: 177.09 (C1), 175.53 (C14), 141.07 (C4), 139.06 (C7), 129.00 (C5), 127.58 (C6), 67.71 (C13), 54.22 (C15), 46.67 (C2), 44.76 (C8), 38.98 (C20), 34.41 (C17), 30.14 (C9), 27.87 (C19), 22.70 (C18), 22.67 (C10), 22.05 (C12), 22.03 (C11), 19.78 (C3). FT-IR: ν (ATR): 2951, 2928, 2867, 2645, 2556, 1736, 1627, 1604, 1528, 1513, 1463, 1453, 1419, 1382, 1358, 1281, 1254, 1228, 1192, 1170, 1146, 1105, 1063, 1023, 986, 924, 897, 881, 849, 819, 791, 754, 709, 682, 631, 655, 598, 544, 530, 523, 475, 443, 418 cm^−1^. UV-Vis (EtOH): λ_max_ = 224.0, 266.0 nm. Elemental analysis: Calc. (%) for C_35_H_56_N_2_O_6_ (600.829): C 69.97, H 9.39, N 4.66, O 15.98, found: C 69.98, H 9.38, N 4.67, O 15.99.

[L-PheOiPr][IBU]—L-phenylalanine isopropyl ester ibuprofenate

L-phenylalanine isopropyl ester ibuprofenate was obtained as a pale yellow solid in 99% yield.



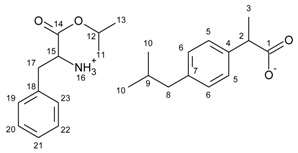



^1^H NMR (400 MHz, CDCl_3_) δ in ppm: 7.30-7.20 (m, 5H, H19, H20, H21, H22, H23), 7.16 (d, J_6,5_ = 6.6 Hz, 2H, H5), 7.08 (d, J_5,6_ = 8.1 Hz, 2H, H6), 5.46 (s, 3H, H16), 5.01 (hept, J_11(13),12_ = 6.3 Hz, 1H, H12), 3.76 (dd, J_17,15_ = 7.2, 5.7 Hz, 1H, H15), 3.66 (q, J_3,2_ = 7.1 Hz, 1H, H2), 3.05 (dd, J_15,17_ = 13.7, 5.7 Hz, 1H, H17), 2.92 (dd, J_15,17_ = 13.7, 7.2 Hz, 1H, H17), 2.43 (d, J_9,8_ = 7.2 Hz, 2H, H8), 1.88–1.78 (m, 1H, H9), 1.47 (d, J_3,2_ = 7.2 Hz, 3H, H3), 1.20 (dd, J_12,11(13)_ = 11.8, 6.3 Hz, 6H, H11, H13), 0.89 (d, J_10,9_ = 6.6 Hz, 6H, H10). ^13^C NMR (100 MHz, CDCl_3_) δ in ppm: 179.10 (C1), 173.73 (C14), 140.49 (C4), 137.96 (C18), 136.54 (C7), 129.42 (C19, C23), 129.15 (C5), 128.62 (C20, C22), 127.27 (C6), 126.97 (C21), 68.84 (C12), 55.09 (C15), 45.08 (C2), 40.21 (C8), 30.20 (C9), 24.86 (C10), 22.43 (C10), 21.78 (C13), 21.71 (C11), 18.39 (C3). FT-IR: ν (ATR): 2977, 2953, 2925, 2867, 2845, 2742, 2653, 2567, 2555, 2550, 2532, 2490, 1747, 1737, 1619, 1570, 1521, 1510, 1495, 1458, 1382, 1365, 1337, 1328, 1290, 1236, 1176, 1106, 743, 699 cm^−1^. UV-Vis (EtOH): λ_max_ = 214.2, 259.0 nm. Elemental analysis: Calc. (%) for C_25_H_35_NO_4_ (413.550): C 72.61, H 8.53, N 3.36, O 15.48, found: C 72.62, H 8.54, N 3.36, O 15.49.

[L-ProOiPr][IBU]—L-proline isopropyl ester ibuprofenate

L- proline isopropyl ester ibuprofenate was obtained as a pale yellow solid in 88% yield.



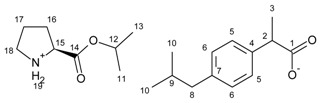



^1^H NMR (400 MHz, CDCl_3_) δ in ppm: 9.13 (s, 2H, H19), 7.23 (d, J_6,5_ = 8.0 Hz, 2H, H5), 7.07 (d, J_5,6_ = 8.1 Hz, 2H, H5), 5.03 (hept, J_11(13),12_ = 6.2 Hz, 1H, H12), 3.99 (dd, J_14,15_ = 8.7, 5.2 Hz, 1H, H15), 3.62 (q, J_3,2_ = 7.2 Hz, 1H, H2), 3.05 (t, J_17,18_ = 6.1 Hz, 2H, H18), 2.43 (d_9,8_, J = 7.1 Hz, 2H, H8), 2.20 (q, J_16(18),17_ = 13.4, 8.0 Hz, 1H, H17), 1.91–1.79 (m, 3H, H16, H17), 1.75–1.66 (m, 1H, H9), 1.45 (dd, J_2.3_ = 7.2, 1.1 Hz, 3H, H3), 1.24 (dd, J_12,11(13)_ = 6.3, 3.3 Hz, 6H, H11, H13), 0.89 (d, J_9,10_ = 6.6 Hz, 6H, H10). ^13^C NMR (100 MHz, CDCl_3_) δ in ppm: 179.28 (C1), 173.00 (C14), 139.90 (C4), 139.48 (C7), 129.15 (C5), 127.25 (C6), 69.49 (C12), 58.49 (C15), 46.28 (C2), 45.88 (C18), 45.10 (C8), 30.20 (C9), 30.02 (C16), 24.86 (C17), 22.44 (C10), 21.70 (C11), 21.69 (C13), 18.90 (C3). FT-IR: ν (ATR): 2986, 2950, 2922, 2867, 1733, 1654, 1647, 1637, 1616, 1576, 1569, 1557, 1511, 1464, 1447, 1381, 1373, 1365, 1354, 1329, 1288, 1230, 1181, 1145, 1097, 1060, 1023, 1002, 945, 905, 871, 854, 813, 799, 726, 685, 543 cm^−1^. UV-Vis (EtOH): λ_max_ = 219.8, 266.0 nm. Elemental analysis: Calc. (%) for C_21_H_33_NO_4_ (363.491): C 69.39, H 9.15, N 3.85, O 17.61, found: C 69.40, H 9.15, N 3.84, O 17.62.

### 3.2. Skin Permeation Studies

The permeation experiments were carried out by using Franz diffusion cells (Phoenix DB-6, ABL&E-JASCO, Wien, Austria) with diffusion areas of 1 cm^2^. The acceptor chamber was 10 cm^3^ and was filled with PBS solution (pH 7.4). In each diffusion unit, there was a constant temperature of 37.0 ± 0.5 °C. The acceptor chamber content was stirred with a magnetic stirring bar at the same speed for all cells. In the experiment, abdominal porcine skin coming from the local slaughterhouse was used. Porcine skin is characterised by similar permeability to human skin [43,44]. The fresh abdominal porcine skin was washed in PBS buffer pH 7.4 several times. Skin of 0.5 mm in thickness was dermatomed. The skin was then divided into 2 cm × 2 cm pieces. The skin samples were wrapped in aluminium foil and stored in a freezer at −20 °C until use, not longer than three months. This frozen storage time was safe for maintaining skin barrier properties [45]. On the day of the experiment, the skin samples were slowly thawed at room temperature for 30 min and were hydrated by PBS pH 7.4 [45,46,47]. The skin was mounted on the donor chamber. Undamaged skin pieces with an even thickness were chosen for experiments. The integrity of the skin was examined by checking its impedance, which was measured using an LCRmeter4080 (Conradelectronic, Germany), which was operated in parallel mode at an alternating frequency of 120 Hz (error at kΩ values < 0.5%). The tips of measuring probes were immersed in the donor and acceptor chamber, filled with PBS (pH 7.4) as described previously [48,49]. Only skin samples of impedance >3 kΩ were applied. These values are similar to the electrical resistance for human skin [49]. The experiment was carried out for 24 h. The samples were reported after 0.5 h, 1 h, 2 h, 3 h, 4 h, 5 h, 8 h, and 24 h of stirring. The donor chamber was filled with 1 cm^3^ of the saturated solutions of tested compounds in PBS (pH 7.4). After this time, the acceptor fluid (0.4 cm^3^) aliquots were withdrawn and refilled with fresh buffer at the same pH. HPLC measured the IBU and its derivative concentrations in the acceptor phase. The cumulative mass (µg IBU·cm^−2^) was calculated based on this concentration. The flux (in µg IBU·cm^−2^·h^−1^) through the pigskin into acceptor fluid was determined as the slope of the plot of cumulative mass in the acceptor fluid versus time.

The accumulation of the tested substance in the skin after penetration was determined using a modification of the methods described by Ossowicz-Rupniewska et al., Janus et al., and Haq and Michniak-Kohl [31,45,50]. After 24 h, the skin samples were removed from the Franz diffusion cell. The skin samples were carefully rinsed in PBS solution at 7.4 pH and dried at room temperature. The skin samples were weighed, cut by diffusion (1 cm^2^), and minced using scissors. Next, skin samples were placed in 2 cm^3^ methanol and were incubated for 24 h at 4 °C. After this time, skin samples were homogenized for 3 min using a homogenizer (IKA^®^T18 digital ULTRA TURRAX (Staufen im Breisgau, Germany)). The homogenate was centrifuged at 3500× *g* rpm for 5 min. The supernatant was collected and analysed using HPLC. Accumulation of the IBU in the skin was calculated by dividing the amount of the drug remaining in the skin by a mass of skin sample and is expressed in mass of ibuprofen per mass of the skin (μg g^−1^).

The concentration of IBU and its derivatives in the acceptor fluid and accumulation in the skin was assessed with a liquid chromatography system (Knauer, Berlin, Germany). The HPLC system consisted of a model 2600 UV detector, Smartline model 1050 pump and Smartline model 3950 autosampler with ClarityChrom 2009 software (Knauer, Berlin, Germany). The detector was operated at 264 nm. A 125 × 4 mm chromatographic column filled with Hyperisil ODS (C18), particle size 5 µm, was used. The mobile phase was 0.02 M potassium dihydrogen phosphate–acetonitrile (60/40 *v*/*v*) with a flow rate of 1 cm^3^ min^−1^. The column temperature was set at 25 °C, and the injection volume was 20 μL.

## 4. Conclusions

Fourteen salts of amino acid isopropyl esters were prepared, described, and characterised. Twelve of these compounds have not yet been described in the literature. The obtained structural modification of ibuprofen significantly increased the solubility of this active compound in both water and PBS aqueous solution (simulating body fluids). Moreover, all the obtained derivatives showed from 2.95 to 16.52 times higher permeability of the active substance as compared with the unmodified ibuprofen. The highest permeability was obtained for L-lysine isopropyl ester monoibuprofenate, which was also characterised by higher transport rates and the greatest mass of permeating ibuprofen. It was also shown that the obtained amino acid derivatives of ibuprofen show from 3.3 to 6.7 times higher accumulation in the skin. The presented results show the possibilities of using the obtained compounds as an alternative to poorly soluble NSAIDs for application to the skin. 

## Figures and Tables

**Figure 1 ijms-23-04158-f001:**
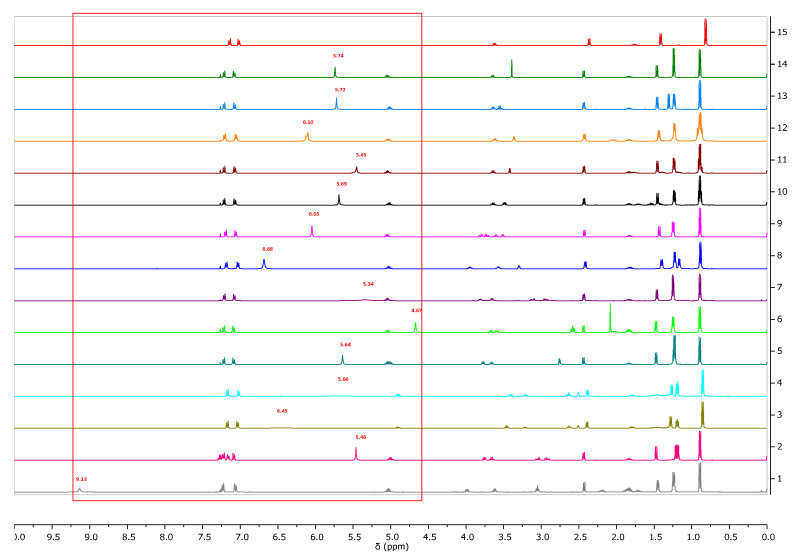
^1^H NMR spectra for ibuprofen and ibuprofen salts, from the top: IBU (red—15), [GlyOiPr][IBU] (green—14), [L-AlaOiPr][IBU] (blue—13), [L-ValOiPr][IBU] (orange—12), [L-IleOiPr][IBU] (maroon—11), [L-LeuOiPr][IBU] (black—10), [L-SerOiPr][IBU] (purplish red—9), [L-ThrOiPr][IBU] (dark blue—8), [L-CysOiPr][IBU] (purple—7), [L-MetOiPr][IBU] (yellow-green—6), [L-Asp(OiPr)_2_][IBU] (cyan-green—5), [L-LysOiPr][IBU] (cyan—4), [L-LysOiPr][IBU]_2_ (yellowish-green—3) [L-PheOiPr][IBU] (pink—2), [L-ProOiPr][IBU] (gray—1) (the amino and protonated amino group are marked in the red square).

**Figure 2 ijms-23-04158-f002:**
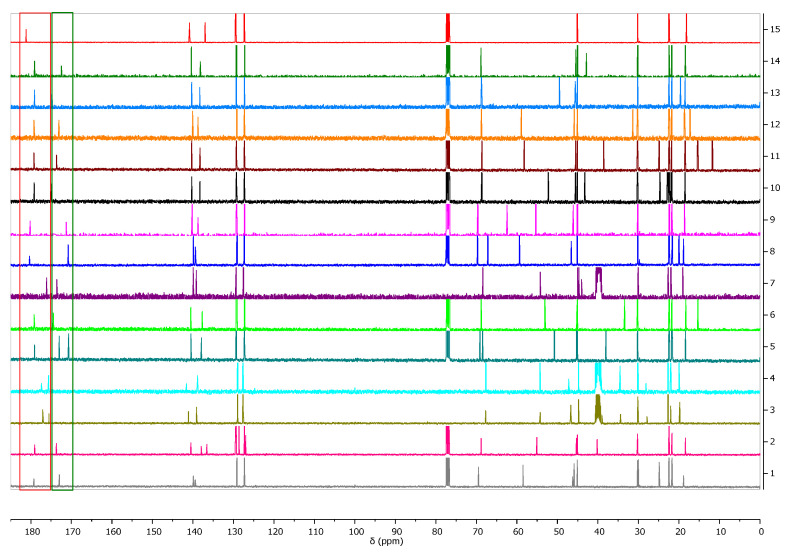
^13^C NMR spectra for ibuprofen and ibuprofen salts, from the top: IBU (red—15), [GlyOiPr][IBU] (green—14), [L-AlaOiPr][IBU] (blue—13), [L-ValOiPr][IBU] (orange—12), [L-IleOiPr][IBU] (maroon—11), [L-LeuOiPr][IBU] (black—10), [L-SerOiPr][IBU] (purplish red—9), [L-ThrOiPr][IBU] (dark blue—8), [L-CysOiPr][IBU] (purple—7), [L-MetOiPr][IBU] (yellow-green—6), [L-Asp(OiPr)_2_][IBU] (cyan-green—5), [L-LysOiPr][IBU] (cyan—4), [L-LysOiPr][IBU]_2_ (yellowish-green—3) [L-PheOiPr][IBU] (pink—2), [L-ProOiPr][IBU] (gray—1) (carbonyl carbons from ibuprofen anions are marked in red square, and carbonyl carbons from amino acid isopropyl esters are marked in green square).

**Figure 3 ijms-23-04158-f003:**
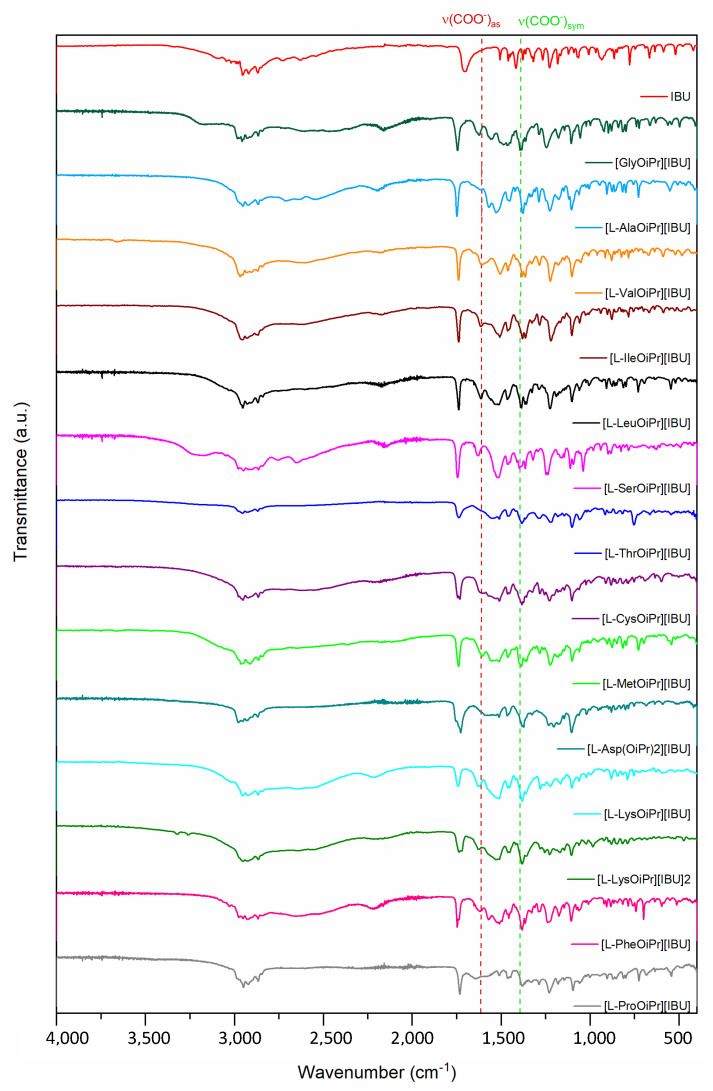
ATR-FTIR for ibuprofen and ibuprofen salts, from the top: IBU (red), [GlyOiPr][IBU] (green), [L-AlaOiPr][IBU] (blue), [L-ValOiPr][IBU] (orange), [L-IleOiPr][IBU] (maroon), [L-LeuOiPr][IBU] (black), [L-SerOiPr][IBU] (purplish red), [L-ThrOiPr][IBU] (dark blue), [L-CysOiPr][IBU] (purple), [L-MetOiPr][IBU] (yellow-green), [L-Asp(OiPr)_2_][IBU] (cyan-green), [L-LysOiPr][IBU] (cyan), [L-LysOiPr][IBU]_2_ (yellowish-green) [L-PheOiPr][IBU] (pink), [L-ProOiPr][IBU] (gray); (ν_sym._) (COO^−^), and asymmetric (ν_asym._) (COO^−^) stretching vibrations marked in green and red, respectively).

**Figure 4 ijms-23-04158-f004:**
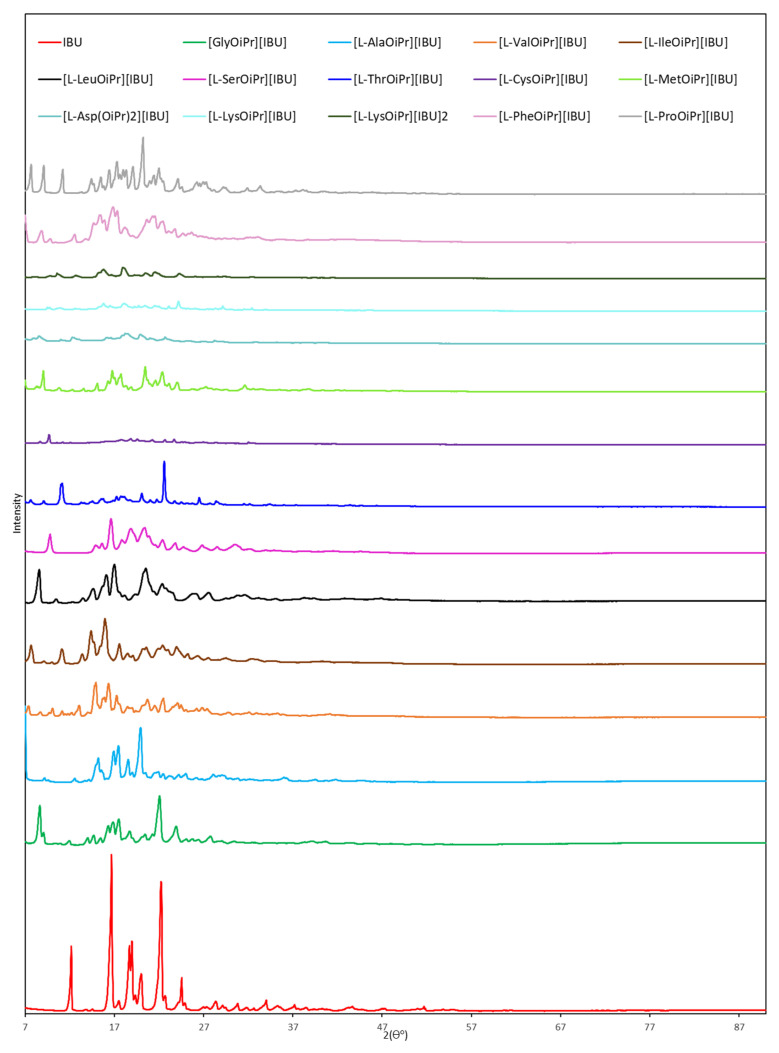
X-ray diffraction patterns of ibuprofen and ibuprofen salts, from the top: IBU (red), [GlyOiPr][IBU] (green), [L-AlaOiPr][IBU] (blue), [L-ValOiPr][IBU] (orange), [L-IleOiPr][IBU] (maroon), [L-LeuOiPr][IBU] (black), [L-SerOiPr][IBU] (purplish red), [L-ThrOiPr][IBU] (dark blue), [L-CysOiPr][IBU] (purple), [L-MetOiPr][IBU] (yellow-green), [L-Asp(OiPr)_2_][IBU] (cyan-green), [L-LysOiPr][IBU] (cyan), [L-LysOiPr][IBU]_2_ (yellowish-green) [L-PheOiPr][IBU] (pink), [L-ProOiPr][IBU] (gray).

**Figure 5 ijms-23-04158-f005:**
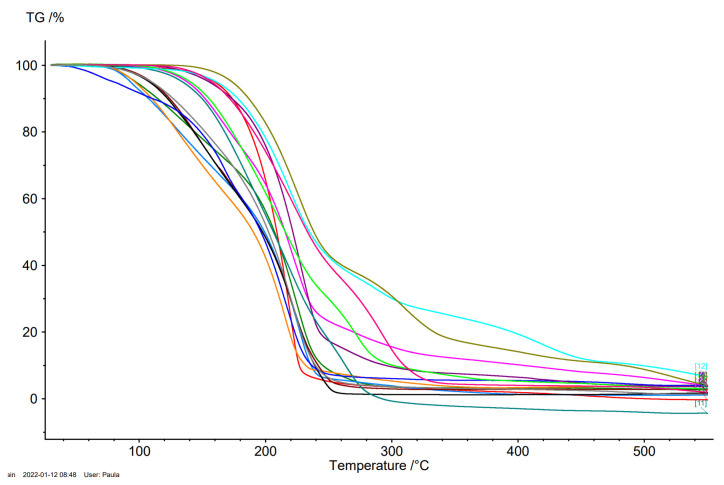
The TG curves of ibuprofen and ibuprofen salts, from the top: IBU (red), [GlyOiPr][IBU] (green), [L-AlaOiPr][IBU] (blue), [L-ValOiPr][IBU] (orange), [L-IleOiPr][IBU] (maroon), [L-LeuOiPr][IBU] (black), [L-SerOiPr][IBU] (purplish red), [L-ThrOiPr][IBU] (dark blue), [L-CysOiPr][IBU] (purple), [L-MetOiPr][IBU] (yellow-green), [L-Asp(OiPr)_2_][IBU] (cyan-green), [L-LysOiPr][IBU] (cyan), [L-LysOiPr][IBU]_2_ (yellowish-green) [L-PheOiPr][IBU] (pink), [L-ProOiPr][IBU] (gray).

**Figure 6 ijms-23-04158-f006:**
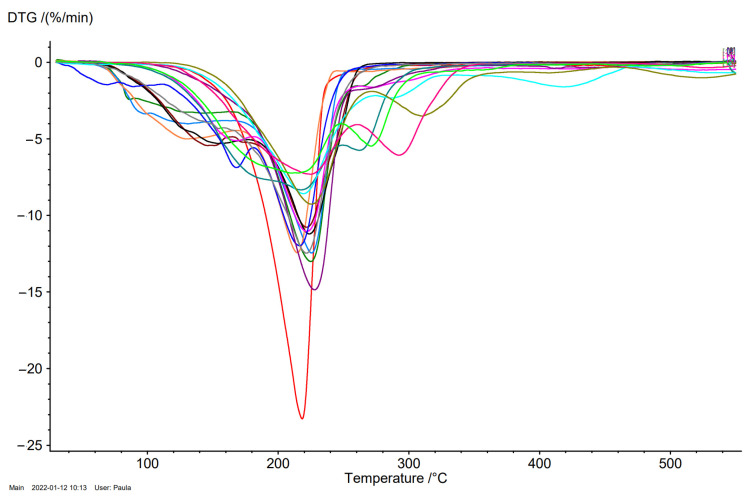
The DTG curves of ibuprofen and ibuprofen salts, from the top: IBU (red), [GlyOiPr][IBU] (green), [L-AlaOiPr][IBU] (blue), [L-ValOiPr][IBU] (orange), [L-IleOiPr][IBU] (maroon), [L-LeuOiPr][IBU] (black), [L-SerOiPr][IBU] (purplish red), [L-ThrOiPr][IBU] (dark blue), [L-CysOiPr][IBU] (purple), [L-MetOiPr][IBU] (yellow-green), [L-Asp(OiPr)_2_][IBU] (cyan-green), [L-LysOiPr][IBU] (cyan), [L-LysOiPr][IBU]_2_ (yellowish-green) [L-PheOiPr][IBU] (pink), [L-ProOiPr][IBU] (gray).

**Figure 7 ijms-23-04158-f007:**
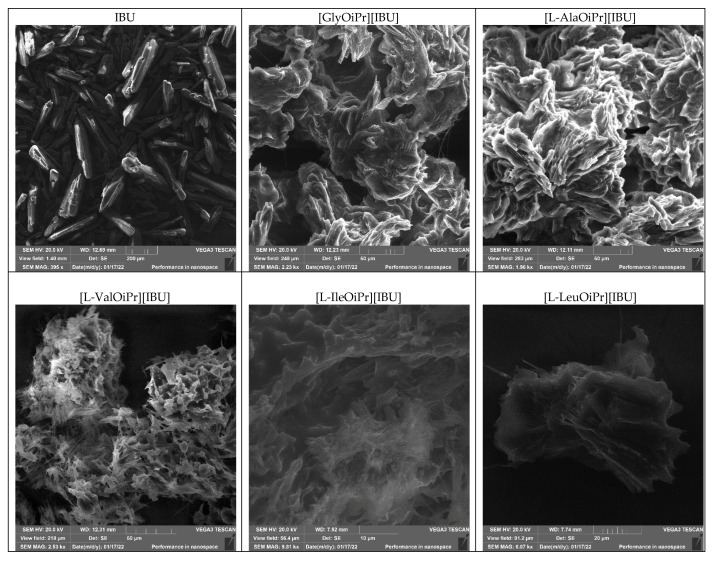
The SEM micrographs of ibuprofen and amino acid isopropyl ester ibuprofenates.

**Figure 8 ijms-23-04158-f008:**
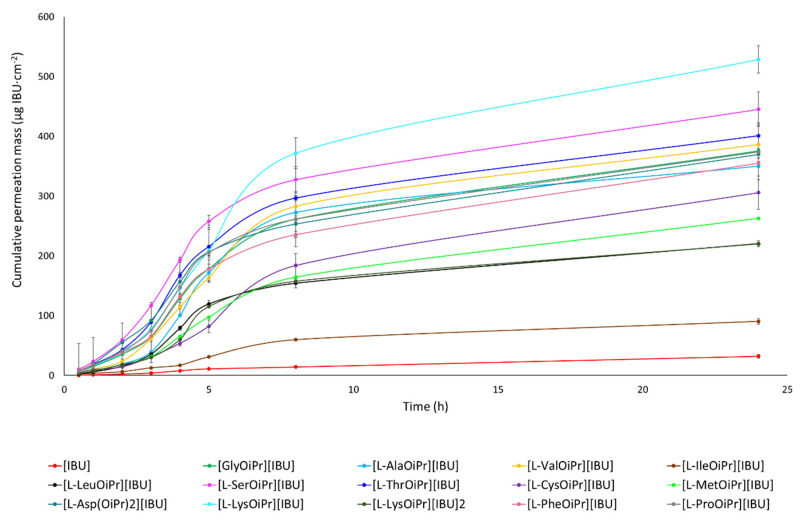
Ibuprofen and amino acid isopropyl ester ibuprofenates permeation profiles. Values are the means with standard deviation; *n* = 3.

**Figure 9 ijms-23-04158-f009:**
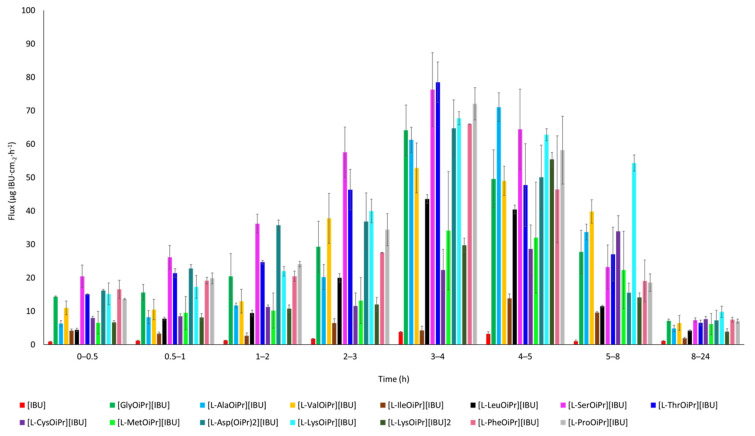
The permeation rate of ibuprofen and amino acid isopropyl ester ibuprofenates during the 24 h permeation; α = 0.05 (mean ± SD, *n* = 3).

**Figure 10 ijms-23-04158-f010:**
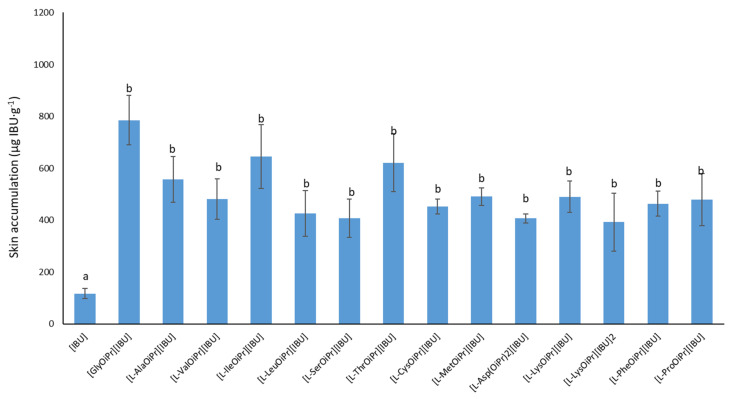
Accumulation in the skin of ibuprofen and amino acid isopropyl ester ibuprofenates during the 24 h penetration; different letters indicate significant differences between the ibuprofenates and control [IBU], *p* < 0.0001, mean ± SD, *n* = 3; (mean ± SD, *n* = 3). The statistically significant difference was estimated by ANOVA using Tukey’s test.

**Figure 11 ijms-23-04158-f011:**
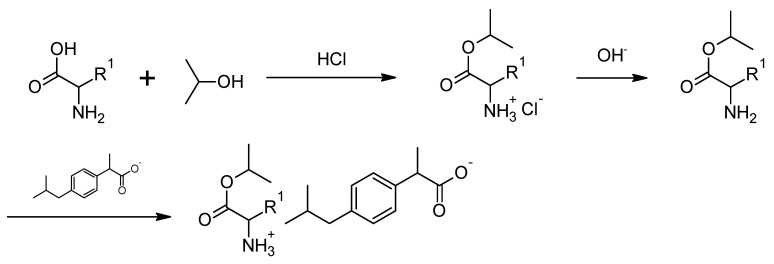
General synthesis path of [AAOiPr][IBU].

**Table 1 ijms-23-04158-t001:** Melting points, thermal stability, and specific and molar rotation for ibuprofen and amino acid isopropyl esters ibuprofenates.

No.	Compound	T_m_ (°C) (DSC)	T_onset_ (°C)	T_max_ (°C)	${\left[\alpha \right]}_{\lambda }^{20}$	${\left[M\right]}_{\lambda }^{20}$
1	IBU	78.6	189.8	218.7	-	-
2	[GlyOiPr][IBU]	85.5	88.7	219.0	+0.393	+1.271
3	[L-AlaOiPr][IBU]	71.0	78.3	221.9	+1.147	+3.871
4	[L-ValOiPr][IBU]	79.0/95.9 [31]	90.2 [31]	215.7	+11.852 [31]	+43.320
5	[L-IleOiPr][IBU]	80.6	105.2	221.9	+15.779	+59.887
6	[L-LeuOiPr][IBU]	88.8	109.5	222.4	+9.162	+34.773
7	[L-SerOiPr][IBU]	94.6	139.2	220.4	−0.794	−2.806
8	[L-ThrOiPr][IBU]	41.3	129.6	209.8	−1.538	−5.654
9	[L-CysOiPr][IBU]	68.1	190.2	229.0	+12.648	+46.736
10	[L-MetOiPr][IBU]	74.0	146.2	209.9	+3.137	+12.472
11	[L-Asp(OiPr)_2_][IBU]	58.4	141.7	217.0	+0.192	+0.881
12	[L-LysOiPr][IBU]	88.3	177.9	219.3	+14.603	+57.616
13	[L-LysOiPr][IBU]_2_	105.3	183.7	225.9	+7.219	+43.374
14	[L-PheOiPr][IBU]	95.4/101.4	175.8	226.6	+13.774	+56.962
15	[L-ProOiPr][IBU]	71.4	128.8	220.5	−19.466	−70.757

T_m_, melting point; T_onset_, the onset temperature of the thermal degradation; ${\left[\alpha \right]}_{\lambda }^{20}$, specific rotation; ${\left[M\right]}_{\lambda }^{20}$, molar rotation.

**Table 2 ijms-23-04158-t002:** Solubility of ibuprofen and amino acid isopropyl ester ibuprofenates in water and organic solvents at 25 °C.

No.	Compound	Ethanol (51.9)	DMSO (45.1)	Dichloromethane (40.7)	Chloroform (39.1)	Ethyl Acetate (38.1)	Diethyl Ether (34.5)	Toluene (33.9)	*n*-hexane (31.0)
1	IBU	+	+	+	+	+	+	+	−
2	[GlyOiPr][IBU]	+	+	+	+	+	+/−	+	−
3	[L-AlaOiPr][IBU]	+	+	+	+	+	+/−	+	−
4	[L-ValOiPr][IBU]	+ [31]	+ [31]	+ [31]	+ [31]	+/− [31]	+/− [31]	+ [31]	− [31]
5	[L-IleOiPr][IBU]	+	+	+	+	+	+/−	+	−
6	[L-LeuOiPr][IBU]	+	+	+	+	+	+/−	+	−
7	[L-SerOiPr][IBU]	+	+	+	+	+/−	−	−	−
8	[L-ThrOiPr][IBU]	+	+	+	+	+/−	+/−	+/−	−
9	[L-CysOiPr][IBU]	+	+	+	+	+/−	-	-	−
10	[L-MetOiPr][IBU]	+	+	+	+	+	+	+	−
11	[L-Asp(OiPr)_2_][IBU]	+	+	+	+	+	+/−	+	−
12	[L-LysOiPr][IBU]	+	+	−	−	−	−	−	−
13	[L-LysOiPr][IBU]_2_	+	+	−	−	−	−	−	−
14	[L-PheOiPr][IBU]	+	+	+	+	+	+/−	+	−
15	[L-ProOiPr][IBU]	+	+	+	+	+	+/−	+	−

Solvents were ranked with decreasing value of empirical solvent polarity parameters, E_T_(30) [41] (“+”: soluble >100 mg cm^−3^; “+/−”: partially soluble 33–100 mg cm^−3^; “−”: insoluble <33 mg cm^−3^) at the temperature 25 °C by modified Vogel’s method [42].

**Table 3 ijms-23-04158-t003:** The solubility of ibuprofen and amino acid isopropyl ester ibuprofenates in water at 25 °C and phosphate buffer pH = 7.4 at 32 °C.

No.	Compound	Solubility in Water		Solubility in Phosphate Buffer pH = 7.4	
		g∙dm^−3^	g IBU∙dm^−3^	g∙dm^−3^	g IBU∙dm^−3^
1	IBU	0.0758 ± 0.001	0.0758 ± 0.001	0.432 ± 0.001	0.432 ± 0.001
2	[GlyOiPr][IBU]	4.580 ± 0.011	2.921 ± 0.007	7.502 ± 0.021	4.785 ± 0.014
3	[L-AlaOiPr][IBU]	4.030 ± 0.014	2.463 ± 0.008	8.440 ± 0.024	5.159 ± 0.015
4	[L-ValOiPr][IBU]	3.468 ± 0.007 [31]	1.957 ± 0.007 [31]	4.998 ± 0.018 [31]	2.821 ± 0.018 [31]
5	[L-IleOiPr][IBU]	2.729 ± 0.180	1.483 ± 0.098	4.713 ± 0.018	2.562 ± 0.010
6	[L-LeuOiPr][IBU]	1.255 ± 0.070	0.682 ± 0.038	3.834 ± 0.003	2.084 ± 0.002
7	[L-SerOiPr][IBU]	4.148 ± 0.009	2.421 ± 0.005	6.929 ± 0.012	4.044 ± 0.007
8	[L-ThrOiPr][IBU]	5.005 ± 0.007	2.809 ± 0.004	6.324 ± 0.004	3.550 ± 0.002
9	[L-CysOiPr][IBU]	2.048 ± 0.253	1.143 ± 0.253	4.437 ± 0.287	2.440 ± 0.287
10	[L-MetOiPr][IBU]	1.191 ± 0.056	0.618 ± 0.029	5.287 ± 0.057	2.743 ± 0.030
11	[L-Asp(OiPr)_2_][IBU]	2.380 ± 0.003	1.159 ± 0.001	2.976 ± 0.018	1.449 ± 0.009
12	[L-LysOiPr][IBU]	9.6534 ± 0.115	5.047 ± 0.060	9.140 ± 0.117	4.779 ± 0.061
13	[L-LysOiPr][IBU]_2_	5.861 ± 0.322	2.012 ± 0.110	6.032 ± 0.342	2.071 ± 0.118
14	[L-PheOiPr][IBU]	0.595 ± 0.048	0.297 ± 0.024	2.039 ± 0.001	1.017 ± 0.001
15	[L-ProOiPr][IBU]	2.594 ± 0.341	1.472 ± 0.194	4.206 ± 0.013	2.387 ± 0.007

**Table 4 ijms-23-04158-t004:** The n-octanol-water partition coefficient (log P) of ibuprofen and amino acid isopropyl ester ibuprofenates.

No.	Compound	Log P
1	IBU	3.208 ± 0.002
2	[GlyOiPr][IBU]	0.645 ± 0.012
3	[L-AlaOiPr][IBU]	0.719 ± 0.004
4	[L-ValOiPr][IBU]	1.154 ± 0.004 [31]
5	[L-IleOiPr][IBU]	1.652 ± 0.008
6	[L-LeuOiPr][IBU]	1.391 ± 0.001
7	[L-SerOiPr][IBU]	0.775 ± 0.008
8	[L-ThrOiPr][IBU]	0.998 ± 0.001
9	[L-CysOiPr][IBU]	1.430 ± 0.005
10	[L-MetOiPr][IBU]	1.509 ± 0.001
11	[L-Asp(OiPr)_2_][IBU]	1.506 ± 0.001
12	[L-LysOiPr][IBU]	1.047 ± 0.018
13	[L-LysOiPr][IBU]_2_	1.055 ± 0.006
14	[L-PheOiPr][IBU]	2.207 ± 0.026
15	[L-ProOiPr][IBU]	1.053 ± 0.048

**Table 5 ijms-23-04158-t005:** Cumulated mass for ibuprofen and amino acid isopropyl ester ibuprofenates; different letters indicate significant differences between amino acid isopropyl ester ibuprofenates and control [IBU], *p* < 0,0001, mean ± SD, *n* = 3. The statistically significant difference was estimated by ANOVA using Tukey’s test.

Compound	Cumulative Permeation Mass	
	(µg·cm^−2^)	(µg IBU·cm^−2^)
[IBU]	32.001 ± 3.167 ^a^	32.001 ± 2.801 ^a^
[GlyOiPr][IBU]	589.273 ± 19.094 ^b^	375.842 ± 12.178 ^b^
[L-AlaOiPr][IBU]	572.719 ± 18.05 ^b^	350.100 ± 11.039 ^b^
[L-ValOiPr][IBU]	684.642 ± 56.109 ^b^	386.391 ± 31.666 ^b^
[L-IleOiPr][IBU]	173.888 ± 8.815 ^b^	94.512 ± 4.791 ^b^
[L-LeuOiPr][IBU]	424.683 ± 9.060 ^b^	230.824 ± 4.924 ^b^
[L-SerOiPr][IBU]	762.855 ± 48.981 ^b^	445.221 ± 20.234 ^b^
[L-ThrOiPr][IBU]	714.248 ± 38.838 ^b^	400.942 ± 17.921 ^b^
[L-CysOiPr][IBU]	541.776 ± 49.664 ^b^	305.756 ± 28.028 ^b^
[L-MetOiPr][IBU]	506.181 ± 23.856 ^b^	262.637 ± 12.378 ^b^
[L-Asp(OiPr)_2_][IBU]	812.658 ± 12.545 ^b^	395.800 ± 41.320 ^b^
[L-LysOiPr][IBU]	1011.106 ± 43.448 ^b^	528.643 ± 22.716 ^b^
[L-LysOiPr][IBU]_2_	640.825 ± 28.577 ^b^	220.016 ± 9.811 ^b^
[L-PheOiPr][IBU]	711.866 ± 5.956 ^b^	355.088 ± 5.482 ^b^
[L-ProOiPr][IBU]	658.661 ± 18.295 ^b^	373.795 ± 10.383 ^b^

**Table 6 ijms-23-04158-t006:** Skin permeation parameters for ibuprofen and amino acid isopropyl ester ibuprofenates; *n* = 3.

No.	Compound	JSS, µg IBU cm^−2^ h^−1^	KP∙103, cm/h	LT, h	D∙104, cm^2^/h	Km	Q%24 h	EF
1	IBU	3.017 ± 0.209	6.984 ± 0.483	1.399 ± 0.145	2.977 ± 0.255	1.173 ± 0.078	7.408 ± 0.648	1.00
2	[GlyOiPr][IBU]	49.305 ± 4.765	10.304 ± 0.996	1.435 ± 0.121	2.903 ± 0.255	1.775 ± 0.317	7.855 ± 0.255	1.06
3	[L-AlaOiPr][IBU]	44.510 ± 0.861	8.627 ± 0.1669	1.691 ± 0.041	2.464 ± 0.061	1.751 ± 0.070	6.786 ± 0.214	0.92
4	[L-ValOiPr][IBU]	43.749 ± 2.445	15.510 ± 0.867	1.443 ± 0.122	2.902 ± 0.256	2.694 ± 0.371	13.698 ± 1.123	1.85
5	[L-IleOiPr][IBU]	9.545 ± 0.239	2.341 ± 0.588	1.617 ± 0.057	2.575 ± 0.093	0.455 ± 0.027	2.318 ± 0.118	0.31
6	[L-LeuOiPr][IBU]	38.502 ± 1.553	18.476 ± 0.745	1.742 ± 0.061	2.394 ± 0.083	3.738 ± 0.278	11.077 ± 0.236	1.50
7	[L-SerOiPr][IBU]	66.106 ± 2.925	16.596 ± 0.723	1.160 ± 0.081	3.556 ± 0.254	2.315 ± 0.256	11.035 ± 0.500	1.49
8	[L-ThrOiPr][IBU]	59.631 ± 6.900	16.798 ± 1.944	1.339 ± 0.177	3.111 ± 0.459	2.699 ± 0.642	11.294 ± 0.505	1.52
9	[L-CysOiPr][IBU]	28.858 ± 2.984	11.524 ± 1.192	1.873 ± 0.104	2.222 ± 0.121	2.590 ± 0.290	12.210 ± 1.119	1.65
10	[L-MetOiPr][IBU]	25.312 ± 0.661	9.227 ± 0.241	1.420 ± 0.054	2.933 ± 0.110	1.573 ± 0.097	9.574 ± 0.451	1.29
11	[L-Asp(OiPr)_2_][IBU]	55.063 ± 1.425	37.989 ± 0.983	1.010 ± 0.054	4.134 ± 0.219	4.607 ± 0.350	27.307 ± 2.851	3.69
12	[L-LysOiPr][IBU]	56.801 ± 4.129	11.886 ± 0.864	1.423 ± 0.130	2.908 ± 0.281	2.044 ± 0.327	11.062 ± 0.475	1.49
13	[L-LysOiPr][IBU]_2_	24.852 ± 2.857	12.000 ± 1.379	1.333 ± 0.258	3.266 ± 0.723	1.919 ± 0.559	10.624 ± 0.474	1.43
14	[L-PheOiPr][IBU]	48.578 ± 4.836	47.762 ± 4.755	1.368 ± 0.091	3.045 ± 0.198	7.843 ± 1.322	35.912 ± 0.539	4.85
15	[L-ProOiPr][IBU]	56.589 ± 2.204	23.708 ± 0.924	1.428 ± 0.031	2.918 ± 0.621	4.065 ± 0.245	15.660 ± 0.435	2.11

J_SS_—steady-state flux; K_P_—permeability coefficient; L_T_—lag time; D—diffusion coefficient; K_m_—skin partition coefficient; Q—the percentage of the applied dose, EF—enhancement factor.

## Data Availability

The data presented in this study are available on request from the corresponding author.

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
