# Peer review of "Influence of the Type of Amino Acid on the Permeability and Properties of Ibuprofenates of Isopropyl Amino Acid Esters"

_ijms, 2022, doi:10.3390/ijms23084158_

Round 1
Reviewer 1 Report
Review
The Manuscript entitled “Influence of the type of amino acid on the permeability and properties of ibuprofenates of isopropyl amino acid esters” is dedicated to the structure modification of ibuprofen with salts of amino acid isopropyl esters followed by the permeability study of obtained compounds.
In order to improve the manuscript, the following suggestions should be taken into account by the authors:
- Please, delete Figure 2-5 and 8 from the body of the paper since these illustrates are presented in Supplementary Material.
- Table 5, Figure 10 and 11 duplicate each other. Therefore, only one graph or Table should be remained.
- The term “Ibuprofenates” does not appear in scientific literature excluding your papers.
- Please, discuss the issue explaining that the charged compounds with lower lipophilicity (synthesized compounds) possess better permeability compared with more lipophilic ibuprofen.
- Please, transfer the procedure of “Skin Permeation Studies” from the Supplementary Material to the body of Manuscript.
- What was the necessity to study the solubility of synthesized compounds in such organic solvents that are presented in Table 2 if further authors investigate the permeability of compounds through the skin?
After this minor revision, I recommend the present manuscript for publication.
Author Response
Dear reviewer,
thank you very much for the effort you put into checking this article. We appreciate your comments very much. We hope that the amendments we have made will gain your approval.
Responses to the review are attached.

Reviewer 2 Report
Dear authors,
A very well written work that provides a huge amount of new information that will be of interest. Please mind my suggestion bellow:
Suggestions/Remarks
- Please provide some indicative literature for introduction paragraph one (i.e. lines 35-47).
- NSAID abbreviation is firstly described correctly in line 37 and thereafter should be provided like that and not reappear in text (e.g. Lines 40-41, line 45, line 61)
- Lines 45-47 what kind of side effects? I believe you should mention more. And take into consideration remark 1.
- Line 52, maybe the authors wanted to say boronylate?
- Lines 55-60, a general figure with the chemical structures will be adding clarity to readership
- Line 63, provide abbreviation OTC
- Line 78, missing value for pH 6.8
- Line 96-98, citations covering the sentences? If it is the same as line 100 please provide as well
- Line 105-106, cite the sentence
- Lines 114-121, same as remark 5…I believe providing a second figure of previous work etc chemical structures at this point will add more.
- Lines 150-155 (figure 2), along with the color assign also the respective numbering found in Y axis! It will add clarity in special occasions
- Lines 165-171 (figure 3), same as remark 11
- Lines 336-338 (figure 10), please provide legend description of blue and green circles as well
- Chapter 3 should also include a link to the SI for the rest methods mentioned therein, because its not clear.
- A chapter 4 is missing, hence conclusions ought to inherit that
Author Response

(The authors gave the same response as above.)
